# Non-allometric expansion and enhanced compartmentalization of Purkinje cell dendrites in the human cerebellum

**Silas E Busch, Christian Hansel***

Department of Neurobiology and Neuroscience Institute, The University of Chicago, Chicago, United States

## eLife Assessment

This is a **convincing** study of the morphological properties of Purkinje cell dendrites and dendritic spines in adult humans and mice, and the anatomical determinants of multi-innervation by climbing fibers. The data will provide an **important** resource for the field of cerebellar computation.

***For correspondence:**
chansel@bsd.uchicago.edu

**Competing interest:** The authors declare that no competing interests exist.

**Abstract** Purkinje cell (PC) dendrites are optimized to integrate the vast cerebellar input array and drive the sole cortical output. PCs are classically seen as stereotypical computational units, yet mouse PCs are morphologically diverse and those with multi-branched structure can receive non-canonical climbing fiber (CF) multi-innervation that confers independent compartment-specific signaling. While otherwise uncharacterized, human PCs are universally multi-branched. Do they exceed allometry to achieve enhanced integrative capacities relative to mouse PCs? To answer this, we used several comparative histology techniques in adult human and mouse to analyze cellular morphology, parallel fiber (PF) and CF input arrangement, and regional PC demographics. Human PCs are substantially larger than previously described; they exceed allometric constraint by cortical thickness and are the largest neuron in the brain with 6–7 cm total dendritic length. Unlike mouse, human PC dendrites ramify horizontally to form a multi-compartment motif that we show can receive multiple CFs. Human spines are denser (6.9 vs 4.9 spines/μm), larger (~0.36 vs 0.29 μm), and include an unreported 'spine cluster' structure—features that may be congruent with enhanced PF association and amplification as human-specific adaptations. By extrapolation, human PCs may receive 500,000 to 1 million synaptic inputs compared with 30–40,000 in mouse. Collectively, human PC morphology and input arrangement is quantitatively and qualitatively distinct from rodent. Multi-branched PCs are more prevalent in posterior and lateral cerebellum, co-varying with functional boundaries, supporting the hypothesis that this morphological motif permits expanded input multiplexing and may subserve task-dependent needs for input association.

## Introduction

Being the sole output of the cerebellar cortex, Purkinje cells (PCs) perform the final step to integrate all cortical input. The dendritic arbor of each PC is the target of tens of thousands of granule cell parallel fiber (PF) axons, which deliver an expansion recoding of mossy fiber contextual input (**Albus, 1971**; **Marr, 1969**) and prediction-error signals carried by one—*or several*—climbing fiber (CF) axons from the inferior olive (**Ito and Kano, 1982**; **Busch and Hansel, 2023**). Understanding cerebellar function thus requires us to understand the process of PC dendritic integration.

PC dendrite structure and function is adapted to receive the most synaptic connections on one of the largest dendritic arbors of the central nervous system. Yet, PC dendrites appear to adapt beyond

being simply large and densely innervated. Recent work has shown that PC dendrites are morphologically diverse (*Busch and Hansel, 2023*; *Kaneko et al., 2011*; *Nedelescu et al., 2018*) and their subbranches can exhibit heterogeneous physiological excitability (*Zang et al., 2018*; *Midtgaard, 1995*; *Ohtsuki, 2020*; *Ohtsuki et al., 2012*; *Roome and Kuhn, 2018*; *Zang and De Schutter, 2021*; *Cirtala and De Schutter, 2024*). Primary dendrite morphology can influence patterns of CF innervation, in some cases permitting non-canonical CF multi-innervation in adult mice that confers independent signaling across compartments in vivo (*Busch and Hansel, 2023*). A study using patch-clamp electrophysiology in cerebellar slices found that dendritic compartments can undergo independent plasticity (*Ohtsuki et al., 2012*). Combined physiological and calcium imaging experiments in awake mice found that intrinsic and synaptic plasticity mechanisms can operate on distinct, compartment-level spatial scales to tune dendritic signaling and gate its impact on axosomatic output (*Lin et al., 2024*).

Morphology thus plays a critical role in shaping PC input integration. While this insight comes largely from the use of the experimentally accessible rodent cerebellum, human PCs are far more enigmatic. Very little is known about the nature of their dendrite morphology, input arrangement, or physiology. Since their illustration by *Golgi, 1874*, and *Ramón y Cajal, 1909*, the morphology of human PC dendrites have only been analyzed occasionally and in small numbers (*Eccles et al., 1967*; *Kato et al., 1985*; *Louis et al., 2014*), eluding systematic quantification. We recently developed a framework for morphological categorization and performed a comparative analysis of thousands of PCs in human and mouse (*Busch and Hansel, 2023*). We defined PCs as having either one ('Normative') or multiple primary dendrite compartments ('Split' when compartments arise from one proximally bifurcated dendrite, or 'Poly' when multiple dendrites emerge directly from the soma). This approach revealed that human PCs are almost universally multi-branched whereas a plurality of mouse PCs are Normative. In mice, ~25% of multi-branched PCs received multiple functionally distinct CFs while Normative cells did not, indicating a link between primary dendrite morphology and cellular physiology. Human PC physiology, on the other hand, remains largely inaccessible, but recent work leveraged limited access to acute human cerebellar tissue and a comparative modeling approach to make important progress (*Masoli et al., 2024*.) To further understand human PC physiology, we will require more complete data on their dendritic morphology and the arrangement of inputs such as PFs and CFs on the arbor.

Here, we provide a comprehensive quantification and comparative analysis of human and mouse PC morphology, excitatory input arrangement, and regional distribution. Our reconstructions reveal that—with dendritic lengths of 6–7 cm—PCs far exceed pyramidal cells in having the largest dendritic arbor in the human brain. They also exceed the allometric constraint of cortical thickness by ramifying horizontally and increasing compartmentalization to produce an expanded multi-branched structural motif. We hypothesized that this motif would permit more input multiplexing and association. In support of this, we found that human cells have increased spine size (~0.36 vs 0.29 µm diameter) and density (6.9 vs 4.9 spines/µm)—receiving upward of 1 million inputs compared with 30–40,000 in mouse—and host a previously unreported 'spine cluster' structure not found in mouse. Co-labeling PCs (calbindin) and CFs (peripherin) confirms that non-canonical CF multi-innervation is present in adult human, as in mouse (*Busch and Hansel, 2023*). The regional prevalence of a multi-branched motif among PC populations covaries with human functional boundaries and exhibits subregional patches of clustered morphologies. This supports the hypothesis that a multi-branched structure can provide for enhanced input multiplexing and that this may subserve complex multimodal association and learning in posterior and lateral cerebellar regions.

## Results

### Human PCs are not an allometrically scaled mouse PC

We used fluorescent calbindin immunohistochemistry in fixed, unembalmed postmortem cerebellar tissue to achieve specific and complete labeling of human PCs for cellular reconstruction (*Figure 1A*; Methods). Unlike embalmed human tissue, as used previously (*Busch and Hansel, 2023*) and for regional analysis below, immunolabeling of unembalmed tissue provides variable sparseness of cell labeling that is ultimately more complete and produces brighter staining that dramatically enhances the signal-to-noise ratio. Calbindin density in mouse tissue precludes individual cell separation, so we achieved sparse labeling via viral expression of Cre-dependent GCaMP6f

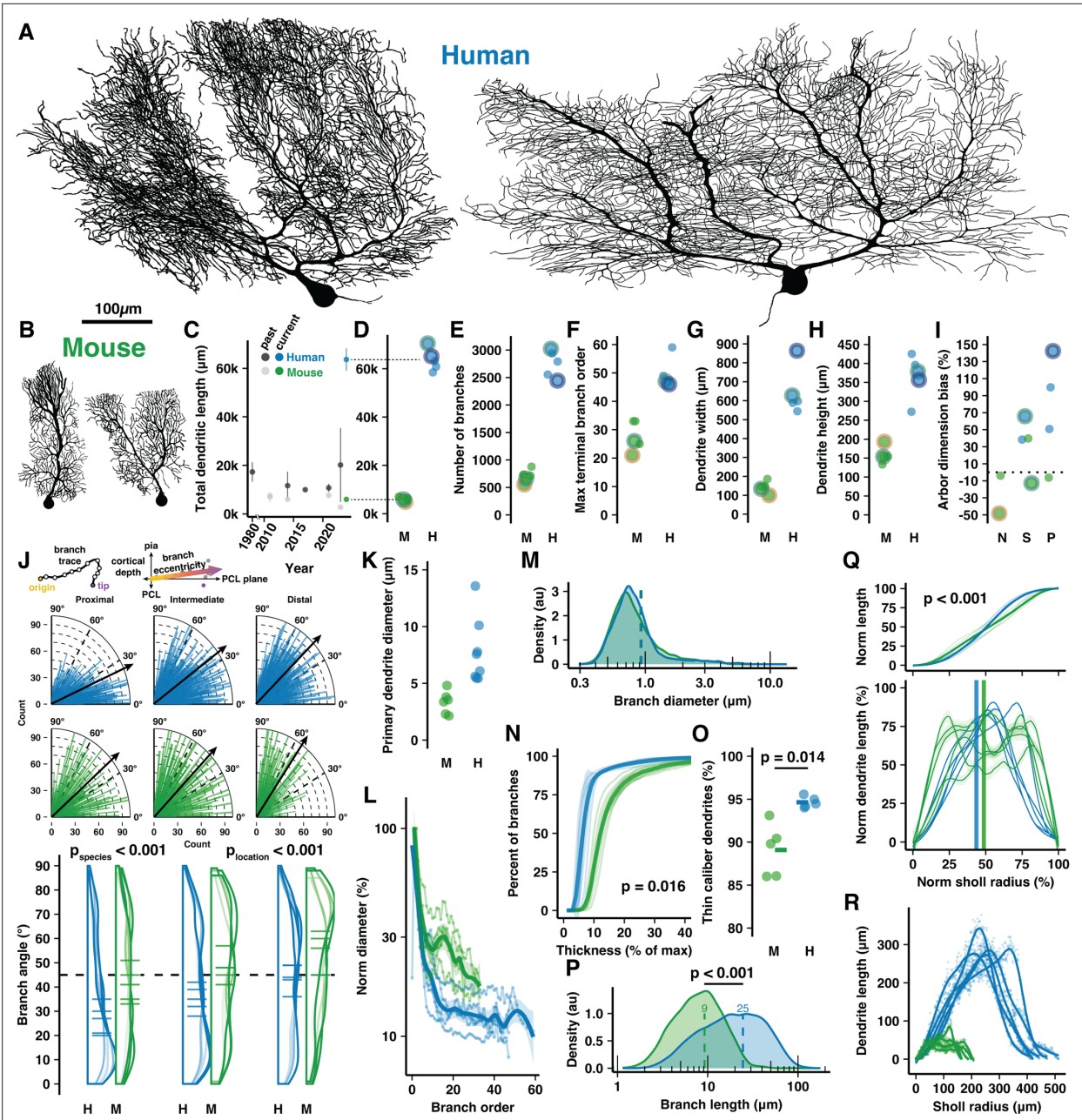

**Figure 1.** Cell reconstructions reveal expanded dendritic size and compartmentalization of human Purkinje cells (PCs). (**A–B**) Manually reconstructed human and mouse PCs (same scale). (**C**) Meta-analysis of historical measures of mean ± SD total dendritic length in human (dark gray) and mouse (light gray) compared with that reported here for human (blue) and mouse (green) (values in D). (**D–I**) Total dendritic length, number of branches, maximum terminal branch order, maximum width and height of the dendritic arbor, and numerical index for the shape of the dendritic area by morphological category (N—Normative, S—Split, P—Poly; positive values indicate arbors with greater width than height; n=5,5 cells, 3,2 individuals). Cells pictured in **A–B** are highlighted by colored outlines according to morphology (Normative = yellow, Split = green, Poly = purple). (**J**) Schematic calculation of branch segment eccentricity relative to cortical depth and the PC layer (PCL) plane (top). Eccentricity data for all branches on quarter-radial plots (middle; black arrows are population mean) and as distributions with cell averages (bottom) by categories of distance from the soma (proximal, intermediate, and distal thirds of ML thickness) in human and mouse (ANOVA, Tukey's HSD post hoc; n=2749,7595,2541 and 877,1247,1056 branches). (**K**) Maximum diameter of the primary dendrite (or dendrites for Poly PCs). (**L**) Dendrite diameter normalized to the thickest primary dendrite by order. (**M**) Distribution of all branch diameters. (**N**) Cumulative distribution of branch segments by their diameter relative to the thickest dendrite (Kolmogorov-Smirnov test; n=5,5 cells). (**O**) Percent of the total dendritic length classified as thin caliber using a 1.31 µm threshold (log-normal mean + 1 SD) of all branch thicknesses across species (Student's t-test; n=5,5 cells). (**P**) Branch segment lengths (Mann-Whitney U test; n=12885,3180 branches). (**Q**) Cumulative (top) and non-cumulative (bottom) normalized distributions of dendritic length by centrifugal Sholl distance from the soma (Kolmogorov-Smirnov test, n=5,5 cells). (**R**) Total dendritic length at each 1 µm Sholl radius without normalization.

*Figure 1 continued on next page*

*Figure 1 continued*

The online version of this article includes the following figure supplement(s) for figure 1:

**Figure supplement 1.** All manual cell reconstructions.

**Figure supplement 2.** Additional morphological data from digital reconstructions.

(*Figure 1B*; Methods). High-resolution z-stack confocal scans (~0.2 μm × 0.2 μm × 0.5 μm voxel resolution) permitted manual reconstruction and analysis of dendritic morphology (n=5,5 cells; *Figure 1—figure supplement 1*). Human PCs were selected for isolation from adjacent cells and completeness of the arbor but minor truncation (approximately <5%) was unavoidable. Occasionally branches from adjacent PCs were visible near and within the area of the dendritic arbor, but these could be reliably excluded from the manual reconstructions as they were disconnected from the cell of interest.

Human PCs were substantially (~11×) longer than in mouse, more than was previously appreciated (*Kaneko et al., 2011*; *Louis et al., 2014*; *Masoli et al., 2024*; *Mavroudis et al., 2017*; *Mavroudis et al., 2022*; *Paula-Barbosa et al., 1980*; *Gao et al., 2011*; *Takeo et al., 2021*; *Joo et al., 2014*; *Figure 1C*), having total dendritic lengths of 63,645±4572 μm and 6004±831 μm, respectively (*Figure 1D*). Human PCs had 3.9× the number of total and terminal branch segments (2750±247 vs 707±118 and 1377±125 vs 355±59; *Figure 1E*, *Figure 1—figure supplement 2A*), which were produced by branching that reached nearly double the maximum and average branch order (50±5.4 vs 28±5.3 and 25±2.5 vs 14±2.6; *Figure 1F*, *Figure 1—figure supplement 2B*). Despite considerable morphological variation, a trend emerged wherein human primary dendrites commonly ramified parallel with the PC layer and bore numerous, often 7–8 (*Figure 1—figure supplement 2C*), compartments that projected vertically toward the pial surface. This alignment rendered their abor ~4.5× wider than in mouse (644±126 μm vs 143±31 μm; *Figure 1G*), while their molecular layer thickness—a crucial allometric variable—defined the maximal arbor height to be only 2.3× taller (366±58 μm vs 158±22 μm; *Figure 1H*). Thus, human PCs attained a horizontal dimensionality with 79±42% greater width than height while mouse PCs had 6.3±31% greater height than width (*Figure 1I*). Horizontal orientation was not limited to the primary dendrites. Across branching orders and somatic distances, human branches ramified at more horizontal eccentricities than in mouse (~37±24° vs 45±24°; *Figure 1J*). Both human and mouse dendrite eccentricities turned upward with increased distance from the soma (~29±23°, 39±24°, 42±23° among branches in proximal, intermediate, and distal compartments vs 41±24°, 43±25°, and 51±23°; *Figure 1J*), a trend conserved across cells (*Figure 1—figure supplement 2E*).

Emerging from a somatic compartment that is 2× the diameter (roughly 7.7× the volume) of mouse (32.8±3.07 μm vs 16.5±0.78 μm; *Figure 1—figure supplement 2D*), human primary dendrites were 2.1× thicker (6.49±2.43 μm vs 3.12±1.17 μm; *Figure 1K*). Human dendrites narrow nearly tenfold as they branch (*Figure 1L*) such that the spiny dendrites of both species converge on an apparently conserved thickness (~0.6–1 μm; *Figure 1M*, *Figure 1—figure supplement 2F*). As a result, most branching orders have only ~15% the diameter of the primary dendrite in human compared with ~25% in mouse (*Figure 1N*). Relatedly, 95% of human dendritic length is devoted to thin, spiny dendrites with <1.3 μm diameter (the log-normal mean+1 SD of dendrite diameter across species) while that figure is 89% in the mouse (*Figure 1O*).

Both species shared patterns of near-symmetric fractal branching over similar relative orders (17/50, 34% vs 10/28, 36%; *Figure 1—figure supplement 2G*) and evenly distributed length (*Figure 1—figure supplement 2H*). Individual branch segments were 2.3× longer on average in human and varied more in their length (24.53±18.78 μm vs 9.23±5.7 μm; *Figure 1P*). The species difference was even greater (3×) among terminal dendritic segments (33.14±19.59 μm vs 10.9±5.81 μm; *Figure 1—figure supplement 2I*). Also reflecting more heterogeneity in the human arbor, the rate of branch emergence in human peaked at precisely half the dendritic height while mouse PC branch segments emerged at a constant rate through roughly the middle 60% of their arbor (*Figure 1—figure supplement 2J*). This pattern was also observed in the distributions of relative total dendritic length (*Figure 1Q–R*) and branch number (*Figure 1—figure supplement 2K–L*) across Sholl radii.

## Human-specific adaptations for size and associative complexity in spine structure and number

As predominantly the sites of excitatory PF input and plasticity, dendritic spines and their morphology contribute to PC physiology. To assess this input pathway, we manually reconstructed ~4000 spines from high-resolution confocal z-stack images (~26 nm × 26 nm × 150 nm voxel resolution; a calculation of the Rayleigh criterion yields a diffraction limit of ~242 nm, see Methods) of spiny dendrite segments across molecular layer compartments of the same tissue as used above for cell reconstruction (*Figure 2A*; Methods). Dendrite sections were chosen for their isolation from nearby branches. This ensured that there was no substantial contamination of the field of view by spinous or dendritic structures from other branches. Some truncated PC dendrite and spine structures were unavoidably present, but these could be distinguished from connected and putatively connected structures when scrolling through the z-plane. Human PCs exhibited higher spine density than mouse PCs (6.81±0.77 vs 5.1±0.61 spines/µm; *Figure 2B*). Both species had higher densities on distal dendrites (6.32±0.62 to 7.1±0.77/µm in proximal vs distal compartments of human; 4.55±0.55 to 5.48±0.35/µm in mouse; *Figure 2C*), suggesting that increased PF input density on distal compartments may be a conserved mammalian trait (*Kano et al., 2018*). Combining measures of total dendritic length and spine density, we can extrapolate that a mouse PC has roughly 30–40,000 spines, consistent with recent findings (*Nguyen et al., 2023*), while a human PC has roughly 400–600,000 spines.

The elevation in human spine density was largely attributable to thin-head spines (5.37±0.82 vs 4.56±0.56/µm) and large mushroom spines (0.49±0.44 vs 0.13±0.14/µm). Thin spine density increased in distal branches of mouse (4.09±0.64 to 4.86±0.28/µm) while mushroom spine density increased in distal branches of human (0.39±0.45 to 0.7±0.59/µm). Both species had similar densities of branched spines (*Harris and Stevens, 1988*; *Lee et al., 2004*; *Lee et al., 2005*; *Loschky et al., 2022*) (0.42±0.18 vs 0.49±0.13/µm) that were elevated on distal branches (0.38±0.19 to 0.45±0.18/µm in human and 0.37±0.11 to 0.6±0.08/µm in mouse).

Head diameters of thin and mushroom spines were relatively larger in human (0.36±0.1 µm vs 0.29±0.08 µm; *Figure 2D*) but note that the absolute values are imperfect as they approach our confocal diffraction limit of 242 nm. As in our cell reconstructions (*Figure 1—figure supplement 2N–O*), spiny dendrite thicknesses were equivalent in both species (0.68±0.1 µm vs 0.73±0.1 µm; *Figure 2—figure supplement 1C*), leading us to hypothesize that human spine necks may be longer to maintain a similar volume ratio between spines and the space surrounding the dendrite (see Methods). Spine necks were indeed longer on average in human (0.83±0.48 µm vs 0.7±0.31 µm; *Figure 2E*), but the ratio of spine to surrounding volume nearly doubled from mouse to human (1.96±0.5% vs 3.73±0.92%; *Figure 2F*), such that elongated necks did not compensate for elevated spine density and size in the human. Increased spine neck length and head diameters made human spines protrude further from the dendrite than in mouse (1.41±0.57 µm vs 1.13±0.41 µm) and with an elevated distance distally (1.34±0.51 µm to 1.47±0.58 µm), while in mice this value was stable (1.16±0.44 µm to 1.13±0.38 µm; *Figure 2—figure supplement 1F*).

In mouse, spines were smaller in distal compartments than proximal (0.28±0.08 vs 0.31±0.08 µm), but in human we observed the opposite effect where distal spine heads were larger than proximal ones (0.43±0.21 vs 0.4±0.17; *Figure 2G*). Again, while the relative values are informative here, the absolute values are imperfect. As a result, the total spine volume fraction around the dendrite remains stable in mouse (1.87±0.24 vs 2.1±0.55%; *Figure 2H*) while in human there is a trend toward higher spine volume on distal branches (4.44±1.36%) than proximal branches (3.52±0.44%). This may indicate that distal compartments amplify synaptic strength through larger spine structure to compensate for increased distance from the somatic compartment.

## Human-specific 'spine cluster' structures throughout the dendritic arbor

The most notable difference between the species was the presence in humans of a previously unreported spine structure we term a 'spine cluster'. Unlike branched spines, spine clusters have one large head with multiple swellings that form distinct punctate structures. Our imaging resolution (Rayleigh limit ~242 nm) clearly precludes a description of their ultrastructure; however, it does permit us to determine that: (1) the puncta emerge from a connected structure rather than being multiple spatially converging spine heads as they are often >400 nm apart, and (2) individual puncta diameters appear

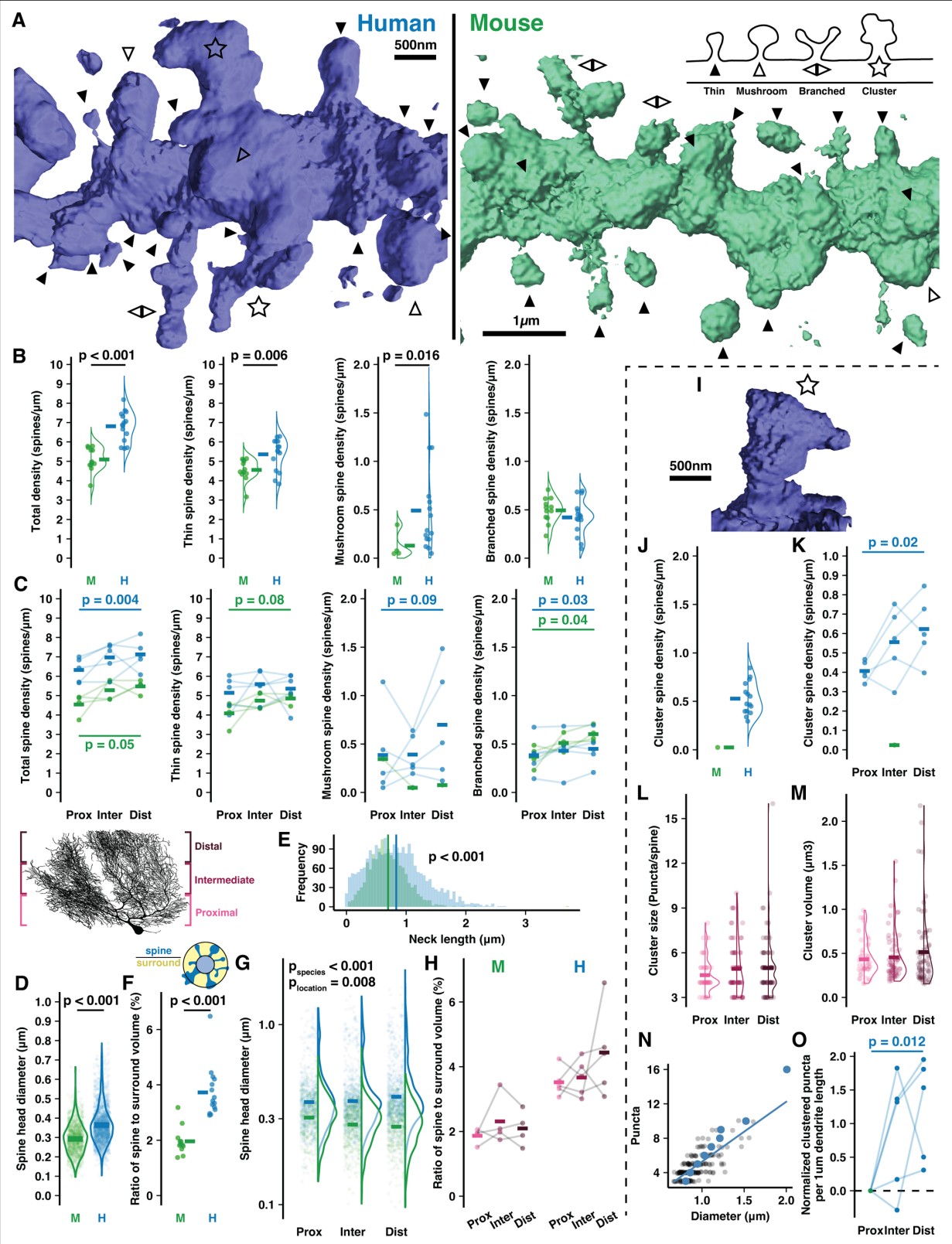

**Figure 2.** Human Purkinje cells (PCs) host expanded sites for putative input and spines with complex morphology. (**A**) Example 3D reconstructions of human and mouse spiny dendrites (background removed for clarity) with inset schematic of spine types: thin (solid arrowhead), mushroom (open arrowhead), branched (back-to-back arrowheads), cluster (star). (**B**) Density of all spines (left) and by spine type in human and mouse (n=15,12 branches; here and below, bars indicate mean). (**C**) Spine densities by branch location relative to the soma (Student's paired one-way t-test for proximal vs distal

*Figure 2 continued on next page*

*Figure 2 continued*

compartments; n=5,4 cells). (**D**) Spine head diameters of thin and mushroom spines (Student's t-test; n=2066,1246 spines). (**E**) Spine neck length by species (Student's t-test; n=2323,1376 spines). (**F**) Ratio of spine head volume to volume surrounding the dendrite (Student's t-test; n=15,12 branches). (**G**) Spine head diameter by branch location (ANOVA, Tukey's HSD post hoc; n=345,478, and 424 spines in human, 676,767, and 812 in mouse). (**H**) Spine to surround ratios by branch location (n=4,4 branches). (**I**) Example spine cluster with enlarged neck diameter and spine head. (**J**) Spine cluster density, included in the total density in (**B**, left). (**K**) Spine cluster density by branch location, included in the total density in (**C**, left) (Student's one-way t-test for proximal vs distal compartments; n=5 individuals). (**L**) Number of puncta per cluster by branch location (n=46,67, and 76 spines). (**M**) Spine cluster head volume by branch location (n=46,67, and 76 spines). (**N**) Spine cluster diameter as a function of puncta number (n=189 spines). Blue points are means by puncta number. (**O**) Total puncta across spine clusters on each branch segment by location and normalized to the proximal branch (Student's one-way t-test for proximal vs distal compartments; n=5 individuals).

The online version of this article includes the following figure supplement(s) for figure 2:

**Figure supplement 1.** Additional data from digitally reconstructed spines.

to be roughly 0.2–0.5 µm. These dimensions resemble thin spine heads and thus perhaps constitute individual synaptic contact sites, although this could not be determined here (*Figure 2*, *Figure 2— figure supplement 1*). While only 1 of 1380 mouse spines met the criteria, spine clusters were present across all human branches, cells, and specimens. It is conceivable that this phenomenon is a factor of aging and not species, as our human specimens came from relatively older individuals (93 and 61yo) than our mouse specimens (12–15 weeks). We addressed this concern by obtaining a tissue sample from the brain of a 37-year-old human (not included in the overall quantitative analysis) and observing the presence of spine clusters there as well (*Figure 2—figure supplement 1B2*). The following analyses focus on the phenomenon in the original, older human samples.

Spine clusters were present at an average density of 0.52±0.16/µm (*Figure 2J*; included in the total of *Figure 2B*) that increased distally from 0.41±0.05 to 0.62±0.17/µm (*Figure 2K*; included in the total of *Figure 2C*). Spine clusters typically had 4–6 puncta (4.84±1.75; *Figure 2L*) and a spherical volume of 0.49±0.42 µm³ (*Figure 2M*) such that the average puncta volume (as a post hoc division of the total) was 0.1 µm³—approximately that of thin spines with 0.3–0.5 µm head diameters. Spine clusters had larger diameters than thin and mushroom spines (0.93±0.19 vs 0.34±0.07 and 0.58±0.07 µm; *Figure 2—figure supplement 1G and H*). The number of puncta, volume, and size of spine clusters were constant across compartments. Diameter and puncta number scale linearly (*Figure 2N*), suggesting that spine clusters vary in size largely through the gain or loss of discrete, stereotypically sized puncta and not through growth or shrinkage of the puncta. The increased number and trend toward increased size of spine clusters in distal compartments contributed to almost double the density of clustered puncta per dendritic length in proximal vs distal compartments (1.85±0.37 to 3.07±0.81/µm; *Figure 2O*).

## CF multi-innervation may be more common in human than mouse cerebellum

While PFs are the predominant source of input onto the spines of thin dendrites, the dendritic shaft and primary dendrite are the sites of a second excitatory projection from CF axons of the inferior olive. CF innervation has been explored extensively in cat (*Eccles et al., 1966*) and rodent models (*Ito and Kano, 1982*; *Maekawa and Simpson, 1972*; *Simpson et al., 1996*) and was recently updated to appreciate multi-innervation of multi-branched PCs in adult mice (*Busch and Hansel, 2023*). During development numerous CF-PC inputs undergo a competitive pruning (*Hashimoto et al., 2009*; *Hashimoto and Kano, 2003*; *Wilson et al., 2019*) that may resemble adult bidirectional synaptic plasticity (*Bosman et al., 2008*; *Hansel and Linden, 2000*; *Ohtsuki and Hirano, 2008*; *Piochon et al., 2016*) and determines which CFs translocate to the dendrite. We previously hypothesized that multi-branched structure offers independent territories for CFs to evade competitive pressure. Ostensibly, being almost exclusively multi-branched may permit a higher prevalence of CF multi-innervation in humans, but we lack physiological or tract-tracing methods to test this, so only a small number of studies *Lin et al., 2014*; *Wu et al., 2021* have addressed adult human CFs after their first depiction by Ramón y Cajal in 1890.

To access this vital input pathway, we immunolabeled CF axonal fibers with the intermediate neurofilament protein peripherin (*Errante et al., 1998*). Though extremely sparse, we identified 44 PCs with co-labeled peripherin fibers that could be classified as: 'putative mono-innervation' (*Figure 3A*),

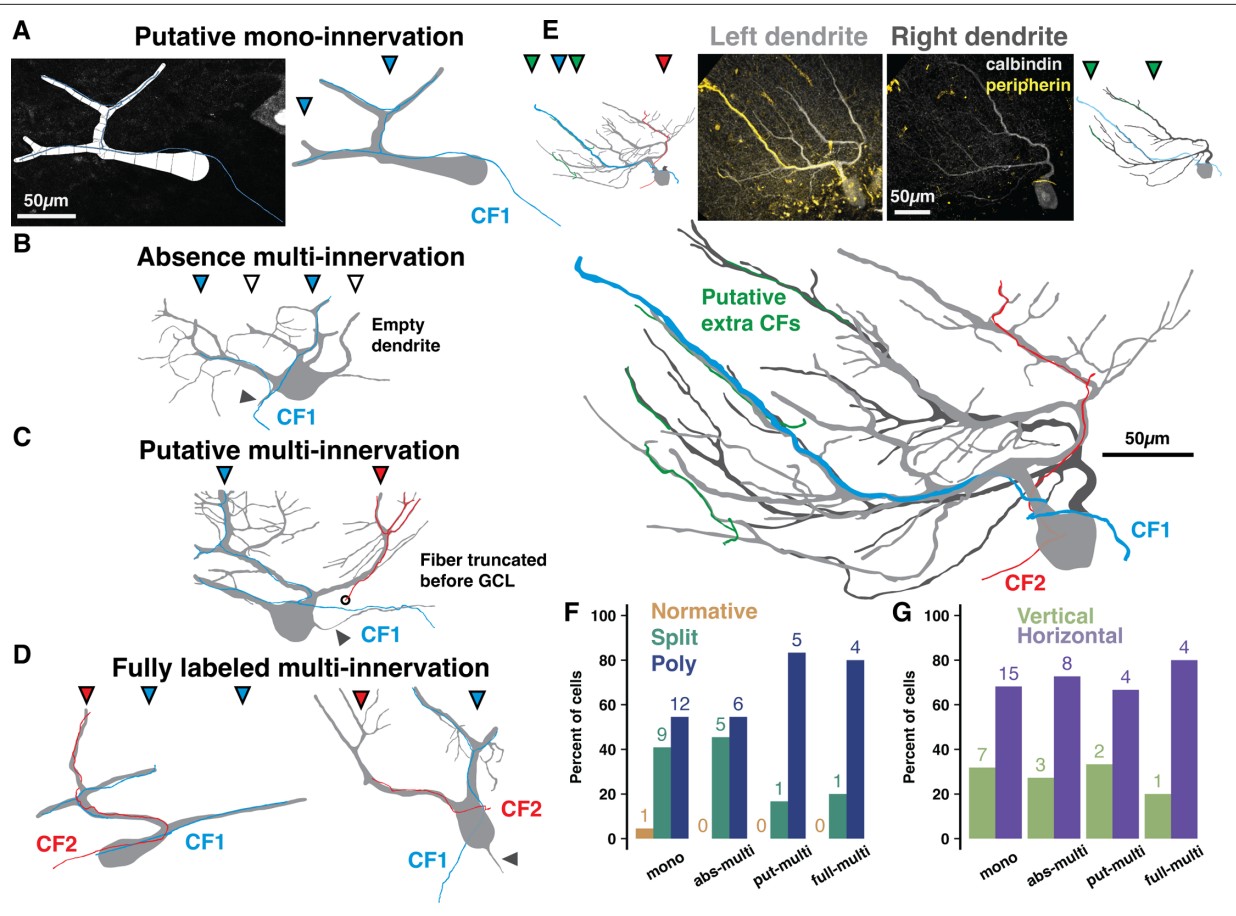

**Figure 3.** Peripherin and calbindin dual-labeling reveals non-canonical climbing fiber (CF) multi-innervation of adult human Purkinje cells (PCs). (**A**) Example reconstruction from human of a PC and peripherin-labeled fiber as originally traced (left) and with masks drawn for visualization (right) to exemplify putative mono-innervation. (**B**) Example PC and peripherin-fiber masks to exemplify absence multi-innervation. Filled and open triangles indicate primary dendrite branches with and without (empty dendrite) a fiber, respectively. Here and below, solid gray triangles indicate the axon. (**C**) Example of putative multi-innervation. Here and below, colors of filled triangles indicate CF identity on each dendrite. (**D**) Examples of fully labeled multi-innervation. (**E**) A Poly PC with multiple peripherin fibers of varying thickness approaching separate dendrite compartments (left dendrite in light gray, right dendrite in dark gray) from distinct locations in the granule cell layer. Untraced composite PC and peripherin-fiber images (top, center) with separate (top, outside) and combined (bottom) masks of each primary dendrite. (**F–G**) Distribution of cell types (**F**) and orientations (**G**) by peripherin fiber classification (n=2 individuals, 44 cells). Numbers above bars indicate absolute counts.

The online version of this article includes the following figure supplement(s) for figure 3:

**Figure supplement 1.** Human Purkinje cells (PCs) with multi-branched and horizontally oriented morphology can avail themselves of additional branch-specific capacities.

incompletely labeled multi-innervation we term either 'absence' and 'putative' multi-innervation (absence: having one fiber that is conspicuously absent from some primary dendrites, *Figure 3B*; putative: having multiple truncated fibers; *Figure 3C*), or 'fully labeled multi-innervation' (*Figure 3D and E*; see Methods). The sparsity precluded analysis of regional or foliar variation; but, taken as an average of posterior hemisphere (L6–8), the cases were classified as: putative mono-innervation (22/44), absence (11/44) or putative multi-innervation (6/44), or fully labeled multi-innervation (5/44). Thus, at least 11% and possibly up to half of human PCs receive multiple CFs, setting the minimum likelihood to the same rate (~15%) of multi-innervation observed in mouse (*Busch and Hansel, 2023*). Variable fiber thickness and arrangement may reflect diverse CF strengths and distributions on the dendritic arbor (*Figure 3E*).

This method naturally underestimates multi-innervation by distant or unlabeled CFs, but it may also be an overestimate as the labeling was not complete enough in the granule cell layer to distinguish extreme cases of 'pseudo-double innervation' by converging branches of the same IO neuron axon

(*Sugihara et al., 1999*). However, the underlying PC morphology offers a modestly more concrete range. In mouse, multi-branched structure and wider dendrite separation distances elevate the rate of multi-innervation (*Busch and Hansel, 2023*). Overall, our peripherin co-labeled PCs have typical rates of multi-branched structure for human posterior hemisphere (2.3% Normative, 36.4% Split, 61.3% Poly). We also categorized co-labeled PCs as having primary dendrite orientations that were either vertical (>30 angle relative to the PCL) or horizontal (parallel with the PCL with a <30 angle; see Methods). Mono-innervated cells, however, are less multi-branched (4.5% Normative, 41% Split, 54.5% Poly; *Figure 3F*) and horizontally oriented (31.8% vertical, 68.2% horizontal; *Figure 3G*) while fully labeled multi-innervated cells are more multi-branched (0% Normative, 20% Split, 80% Poly) and horizontal (20% vertical, 80% horizontal). Factors shaping the relationship between multi-branched structure and CF multi-innervation may thus be conserved across mice and humans. The demographics of intermediate classifications diverge: absence multi-innervation PCs (0% Normative, 45.5% Split, 54.5% Poly) closely align with mono-innervated PCs while putative multi-innervation PCs (0% Normative, 16.7% Split, 83.3% Poly) resemble fully labeled multi-innervated PCs (*Figure 3F*). Thus, putative and fully labeled cases may in fact demarcate the same phenomenon, multi-innervation, and set the population rate to at least 25% in human.

Beyond CFs, our staining reveals that segregated primary dendrite compartments can be the recipient (*Figure 3—figure supplement 1A*) of narrowly branching recurrent PC axon terminals (*Ramón y Cajal, 1909*). In mouse, recurrent PC axons target interneurons, PC somata (*Witter et al., 2016*), and superficial granule cells (*Guo et al., 2016*) on rare occasion a diminutive branch will ascend into the molecular layer. We also found that primary dendrites sometimes host the axon, thereby differentiating the axon initiation site exposure to the distinct primary dendrite signals (*Figure 3—figure supplement 1B*). This has not been shown in cerebellum of any species and complements our finding that PC output may be disproportionately driven by subsets of input.

## Regional distributions of PC demographics in vermis are distinct from hemisphere and align with human functional boundaries

Because PC dendritic computation and input arrangement is likely shaped by morphology, we next asked whether PC morphology demographics vary across cerebellar regions in alignment with local task-specific demands. The prevalence of multi-branched PCs increases in more posterior hemisphere lobules of both species, forming an anterior-posterior gradient (*Busch and Hansel, 2023*). This may reflect that multi-branched morphology—and the input multiplexing and independent branch signaling this confers—is needed for multi-modal associative tasks performed by posterior hemisphere. If PC demographics indeed align with functional demands, we hypothesized that the cerebellar vermis, having more evolutionarily conserved somatosensory and motor functions like anterior hemisphere (*Nettekoven et al., 2024*; *Buckner et al., 2011*; *Saadon-Grosman et al., 2022*), would lack this gradient and demographically resemble the anterior hemisphere.

To test this, we used our previous framework to categorize PCs by their primary dendrite morphology (Normative, Split, Poly) and orientation (Vertical or Horizontal) in the vermis of human and mouse. This could be done exhaustively due to the small size of the mouse or the more sparse labeling of embalmed human tissue as compared with unembalmed tissue from previous experiments (*Figure 4A*, *Figure 4—figure supplement 1A and B*). Unlike the previous cell, spine, and fiber reconstructions in unembalmed tissue, the specimens used here were the same embalmed individuals as in our previous work (*Busch and Hansel, 2023*). This way, all regions can be directly compared within individual. Indeed, vermis lobules lacked a clear anterior-posterior gradient and largely resembled anterior hemisphere with an elevated rate of normative PCs and more Split than Poly PCs (*Figure 4B*, bottom), though vermis PC orientations (*Figure 4B*, top) better resembled posterior hemisphere (*Figure 6—figure supplement 1A–C*). These regional trends were similar in the mouse (*Figure 4C and D*). In human, the anterior- and posterior-most lobules (L1–2 and L9–10) diverged from the general trend by having higher rates of Poly, Normative, and vertically oriented PCs. A similar divergence was observed in the anterior-most but not posterior-most regions of the mouse. Instead, mouse L7–9 appeared distinct in having a higher rate of Poly PCs.

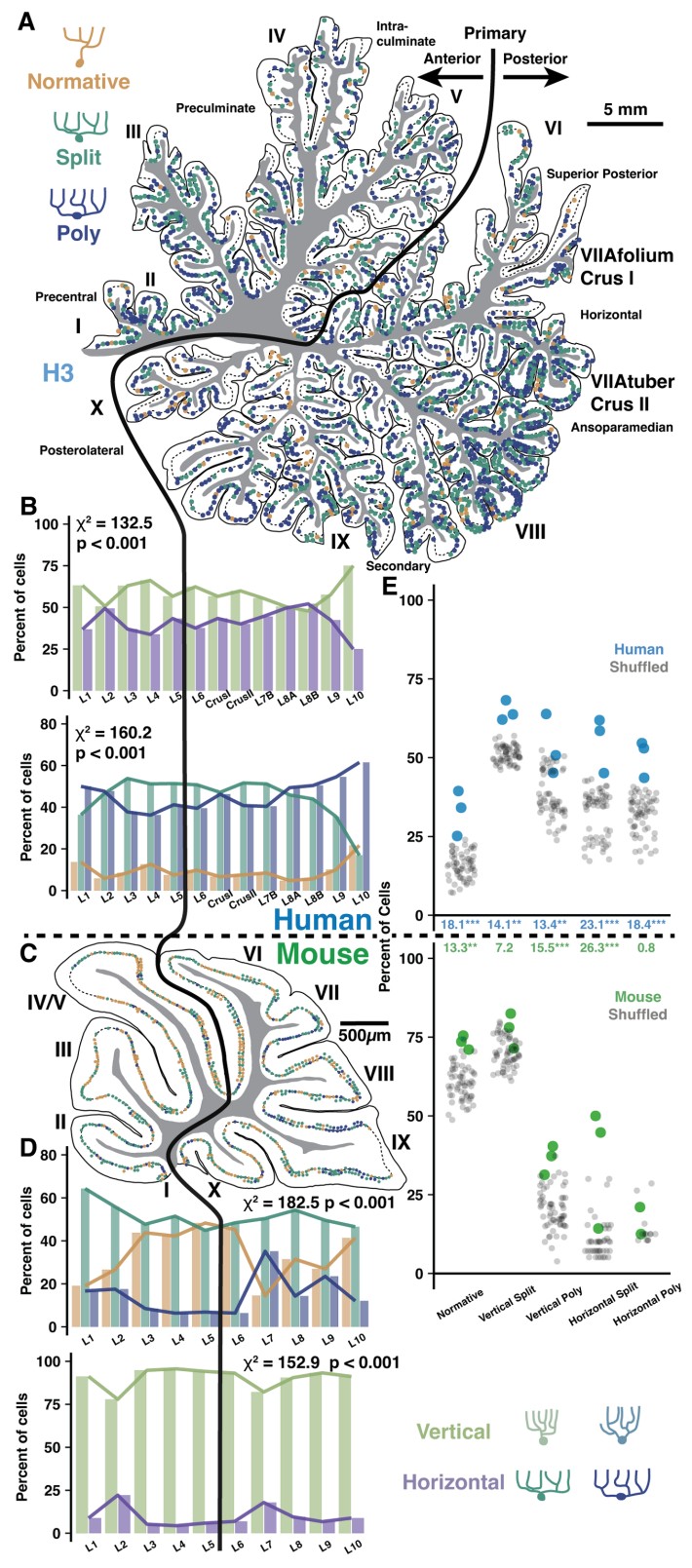

**Figure 4.** Regional and locally clustered Purkinje cell (PC) demographics in the vermis of human and mouse. (**A**) Example exhaustive reconstruction of PC morphological distributions in a parasagittal section of human vermis. (**B**) Morphological orientation (top) and type (bottom) demographics across lobules (chi-squared test; n=3 individuals, 6346 cells). (**C**) Example exhaustive reconstruction of a parasagittal section from mouse vermis. (**D**) Morphological

*Figure 4 continued on next page*

*Figure 4 continued*

type (top) and orientation (bottom) demographics across lobules in mouse (chi-squared test; n=3 individuals, 2284 cells). (**E**) Observed and shuffled rates of adjacent PC clustering in human and mouse (within species ANOVA, Tukey's HSD post hoc; n=3 individuals and 20 shuffles of each). Numbers between graphs indicate the difference between observed and shuffled mean. *p<0.05, **p<0.01, ***p<0.005.

The online version of this article includes the following figure supplement(s) for figure 4:

**Figure supplement 1.** Additional reconstructions and analysis of Purkinje cell (PC) demographics in parasagittal slices of vermis from human and mouse.

## Local cell-type clustering produces 'patchy' heterogenous processing zones within regions

Human fMRI studies demonstrate that functional boundaries do not align with the anatomical boundaries of lobules. Particularly given the size and foliation of human cerebellum, it is not clear at what spatial scale PC demographics are compiled to serve computational needs. We asked whether task-specific computational demands act on PC populations at the level of whole regions, individual folia, or smaller circuits of neighboring cells with overlapping dendrites. The distribution of PC morphologies would thus sort randomly throughout a region, be homogeneous in each folium, or exhibit 'patches' of homogeneous morphologies among neighboring PCs.

To assess demographic sorting, we first tested cell-type clustering at the scale of adjacent PCs close enough to partially overlap their dendrites (*Figure 5*). Each cell received a score between 0 and 2 to reflect how many adjacent cells matched their morphological type (Normative, Split, or Poly) and orientation (Vertical or Horizontal) along the parasagittal line of the PC layer (*Figure 5A*). The rate of nonzero cluster scores was compared to a shuffled dataset (see Methods), revealing that adjacent cells with possible dendritic overlap were more likely than chance to share morphological categories (*Figure 4E*). The nonrandom clustering rate was modestly more robust in human than mouse (17.4 vs 12.6%). Clustering was only apparent when requiring a complete match of morphological type and orientation (*Figure 5B and C*), not for partial matches (*Figure 5—figure supplement 1C and D*), in both vermis and hemisphere (*Figure 4—figure supplement 1F and G*, *Figure 6—figure supplement 1F and G*). Conceivably, clustering may produce even more precise morphological similarity among adjacent cells than surmised via broad categories. This analysis also revealed that uninterrupted clusters have on average 2.6 cells forming an inter-somatic length of 537 µm in human vermis (*Figure 5B and C*). In mouse, uninterrupted clusters have 3.1 cells forming a 71 µm length. Compared with clusters in the vermis, clusters in the hemisphere were longer in human (2.7 cells and 838 µm length; *Figure 5—figure supplement 1E and F*) but identical in mouse (3 cells with 70 µm length).

PC shapes are often congruent with the expansion of the molecular layer in the gyral lip or the compression in the base of the sulcus (*Eccles et al., 1967*; *Friede, 1955*). While this suggests that clustering may be mechanically driven by tissue foliation, we find little variation in PC demographics across foliar location (gyrus, bank, or sulcus) in human and mouse (*Figure 4—figure supplement 1C*), though there were more horizontally oriented PCs in human sulcus (*Figure 4—figure supplement 1D*). The same patterns were observed in hemisphere (*Busch and Hansel, 2023*). To control for foliation, we performed the same clustering analyses with shuffled data where foliar location was held constant (see Methods). Nonrandom clustering was still observed (*Figure 4—figure supplement 1E*), particularly in human vermis, but also in mouse and human hemisphere (*Figure 6—figure supplement 1D and E*). Separating by foliar sub-area, clustering was present within bank regions—lacking bias from mechanical force—as well as the gyrus and sulcus of both vermis (*Figure 4—figure supplement 1F and G*) and hemisphere (*Figure 6—figure supplement 1F and G*).

It is also possible that clustering occurs among a broad population where most, but not all, cells are the same. The previous analysis could not resolve this, so we measured how well the morphology of each cell was matched by the demographics of all surrounding cells within variable distances (*Figure 5D*). Nonrandom clustering provided an increased chance that surrounding cells had a matching morphology by ~13–14% in human and ~9–10% in mouse. Nonmatching cells, inversely, had diminished likelihoods. The elevated prevalence fell below 5% within 2 mm and 500 µm in human and mouse, respectively, and dropped to near zero at scales roughly over 1 cm and 2 mm. Clustering within the core of a measured area could artificially inflate the rate of drop-off, so we next sampled only cells surrounding variably

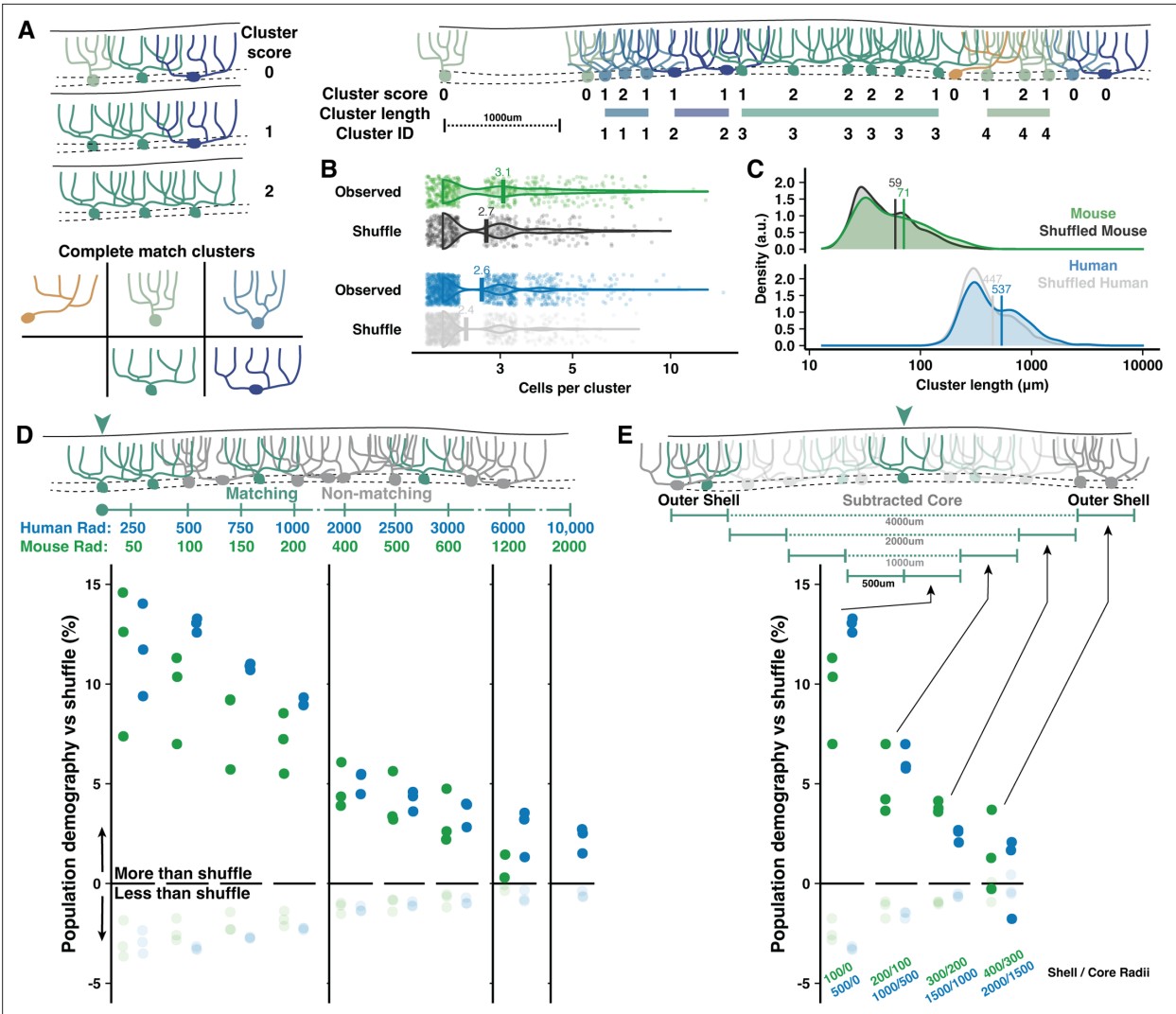

**Figure 5.** Adjacent and local nonrandom cell-type clustering in human and mouse vermis. (**A**) Schematic of cluster score assignment based on complete morphological match (left) and score use to identify the number of cells and parasagittal length of clusters (right). (**B–C**) Number of cells (**B**) and length (**C**) of Purkinje cell (PC) clusters in mouse and human compared to shuffled data (Mann-Whitney U test; n=288,1295 observed clusters; n=263,1122 shuffled clusters). (**D**) Schematic of local population demographics measured over variable distances (top). Radii in mouse are 20% the length in human, matching the difference in dendritic width. Rates of matching (dark points) and nonmatching (light points) morphologies in a shuffled population are subtracted from unshuffled data to measure the percent elevation of clustering in human and mouse. Points mark the average demographic difference between observed and shuffled rates across all five cell groups (i.e. one dark point represents the average for matching cell rates of Normative to Normative, Split to Split, etc.). (**E**) To measure the absolute spatial scale of elevated PC clustering, we measure the observed vs shuffled rate for 500 μm increments of a shell region around a growing core. By ignoring the morphologies of the core region, we exclude local clustering from measurements of distant clustering.

The online version of this article includes the following figure supplement(s) for figure 5:

**Figure supplement 1.** Adjacent Purkinje cell (PC) demographics reveal the spatial scale of nonrandom morphological clustering among local PCs.

**Figure supplement 2.** Population Purkinje cell (PC) demographics reveal the spatial scale of nonrandom morphological clustering among parasagittal PC circuits.

sized core regions (*Figure 5E*). This revealed that clustering was largely restricted to a 2 mm and 400 μm range in human and mouse, respectively. The scale of nonrandom clustering in the hemisphere extended to 3 mm in human and held at 400 μm in mouse (*Figure 5—figure supplement 2*).

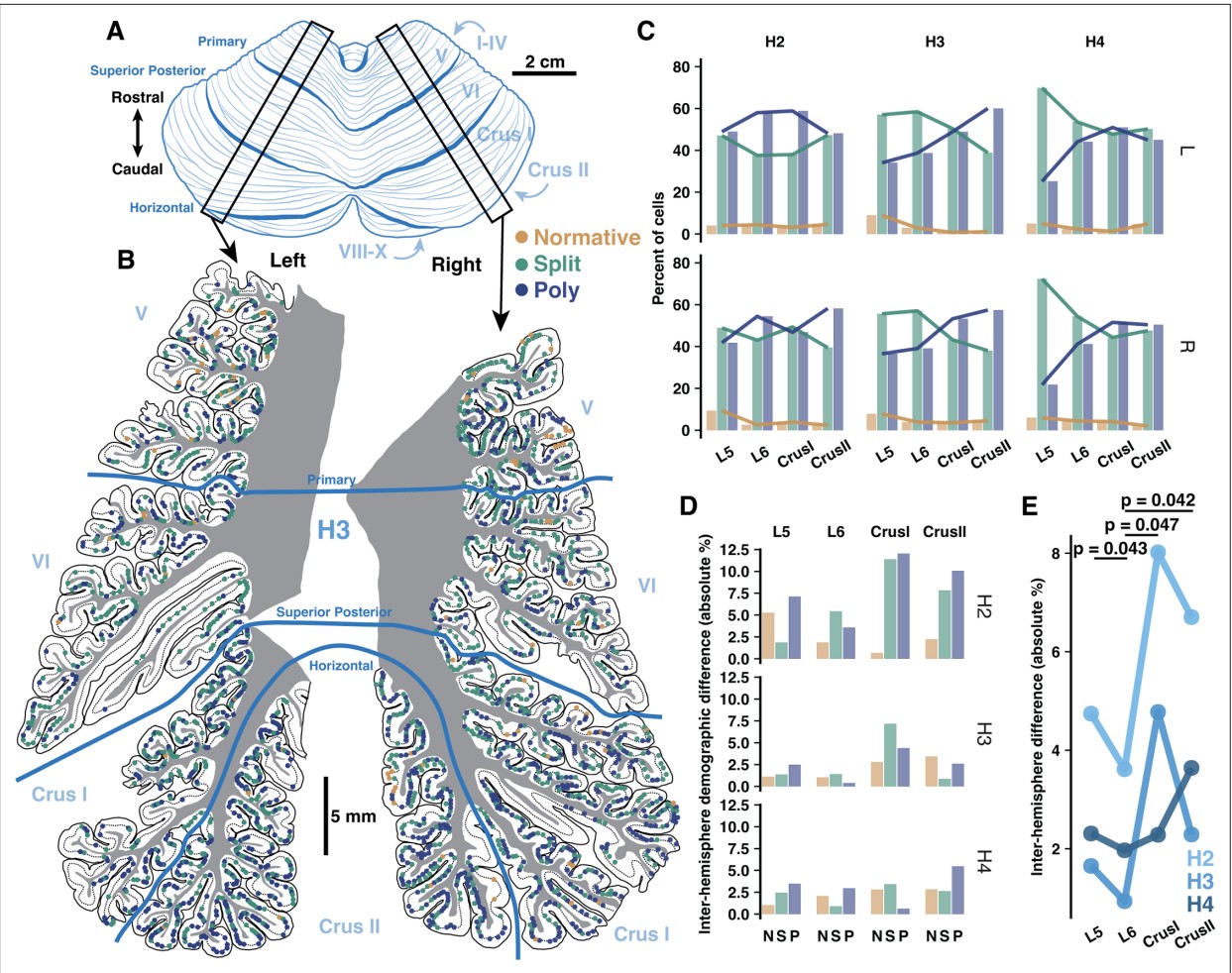

**Figure 6.** Inter-hemisphere similarity of Purkinje cell (PC) demographics is congruent with functional lateralization. (**A**) Schematic of human cerebellum. (**B**) Parasagittal reconstructions of PC morphological distributions in left and right mid-hemisphere lobules L5-Crus II within the same individual. (**C**) Morphological demographics across lobules in left (top) and right (bottom) hemispheres by individual. (**D**) Absolute differences of morphological demographics across hemispheres by lobule and individual. (**E**) Normalized mean inter-hemisphere demographic difference from L6 by lobule and individual (Student's one-way t-test; n=3 individuals).

The online version of this article includes the following figure supplement(s) for figure 6:

**Figure supplement 1.** Additional reconstructions and analysis of Purkinje cell (PC) demographics in parasagittal slices of the mid-hemisphere of human.

**Figure supplement 2.** Human cerebellar regional borders as defined by morphological demographics or functional imaging.

## Inter-hemisphere PC demographic similarity covaries with functional symmetry

As some lobules have higher functional asymmetry across hemispheres in human (*Nettekoven et al., 2024*; *Stoodley and Schmahmann, 2009*; *Wang et al., 2013*; *Wang et al., 2014*; *O'Reilly et al., 2010*), we hypothesized that inter-hemisphere PC demography might reflect these patterns if it is related to function. Functionally symmetric regions (e.g. L5–6) may then exhibit more similar PC demographics than highly asymmetric regions (e.g. Crus I-II). To test this, we compared PC demographics in lobules 5-Crus II of the opposing hemisphere of specimens from which we previously examined one hemisphere (*Busch and Hansel, 2023*; *Figure 6A and B*, *Figure 6—figure supplement 1A and B*). We now show that the previously described anterior-posterior gradient is a *bilateral* phenomenon, as Poly PCs were universally more prevalent in Crus I/II than L5–6 (*Figure 6C*). The inter-hemisphere rate of PC subtypes varied more in Crus I/II than L5 and especially L6 (*Figure 6D*). Averaging the difference across morphologies (*Figure 6E*), the mean inter-hemisphere demographic variation of each lobule confirmed that L5 and L6 exhibited less variation (2.9±1.63% and 2.17±1.35%) than Crus I

and II (5.03±2.88% and 4.21±2.26%). Due to high inter-individual variation, we normalized the rate of variation in L5, Crus I, and Crus II to that of L6 within individuals. This revealed that all of those lobules had more variation than L6.

## Discussion

Our histological interrogation of PCs in postmortem human and mouse tissue defines human PC morphology and input arrangement as both *quantitatively* and *qualitatively* distinct from the rodent model. Recent work using developmental and spatial transcriptomics in human tissue has revealed that PCs are genetically diverse (*Sepp et al., 2024*; *Zhong et al., 2023*; *Aldinger et al., 2021*; *Kozareva et al., 2021*). Conservative assessments identify four developmental PC subtypes. While these four subtypes are equally identified in human and mouse PCs, the subtype ratios shifted in mammalian evolution (*Sepp et al., 2024*). Moreover, human-specific PC genes have been suggested (*Buchholz et al., 2020*). Thus, variable genetic profiles may factor into the diversification of PC morphology within species and may also underlie the human-specific features observed here. In clarifying how the species diverge, this work not only highlights the importance of further anatomical study in humans but also strengthens the interpretability of disease studies that must by necessity be pursued in animal and in silico models (*Masoli et al., 2024*). Dendrite morphology may factor into cerebellar disease etiology. In mouse, PC morphology and CF signaling is altered in disease models such as spinocerebellar ataxia (*Du et al., 2013*; *Du et al., 2019*) and syndromic autism (*Simmons et al., 2022*; *Busch et al., 2023*; *Piochon et al., 2014*). On the only occasions that human PC dendritic morphology has been measured with respect to cerebellar disease, it was disrupted in essential tremor (*Louis et al., 2014*; *Mavroudis et al., 2022*) and Friedreich's ataxia (*Kemp et al., 2016*), and dendritic abnormalities were recently reported in spinocerebellar ataxias and other neurodegenerative movement disorders (*Louis et al., 2023*).

### Are human PCs an allometrically scaled mouse PC?

Human PCs are substantially larger than those of the mouse. But are they *just* allometrically scaled mouse PCs? The answer is likely no. Only some features appear to be either invariant or proportionally scaled with molecular layer thickness. We observed similar fractal branching patterns and relative average distances of peak branch number and dendritic length from the soma, suggesting a possible shared dependence on cell-autonomous factors governing the branching pattern. The diameter of thin caliber dendrites is identical, indicating a shared constraint on dendritic thickness, perhaps by limits of physiological conduction.

The dimensions of the cerebellar cortex are a plausible allometric constraint on cellular growth. Yet, human PCs expand their size well beyond the relative difference in molecular layer thickness—reflected by PC height—by multiplying and extending their primary dendrites outward in the parasagittal axis to articulate an expanded horizontal profile. The horizontal eccentricities of non-primary dendrites further drive the extended width of human PCs and permit spiny branches to double back, overlap in distinct mediolateral planes, and possibly resurvey the PF axons projecting through the arbor. The latter feature could expand combinatorial coding by human PC dendritic branches without requiring increased granule cell or PF densities. Human PCs also exhibit more variable morphological features across locations in the arbor, such as having a rapid rise and decline in branch location, total length, and branching order relative to the position of the soma. The complexity of this distribution is distinct from mouse PCs where these properties are maintained homogeneously through most of their structure. More variable branch lengths may also diversify independent signaling capacity across the arbor. While this cannot currently be tested directly, the confluence of invariant, allometric, and non-allometric morphological features may indicate that human PC physiology diverges from the rodent not only as a matter of scale (*Masoli et al., 2024*) but as a matter of kind.

### Enhanced and human-specific features of input amplification and association by dendritic spines

*Qualitative*, unlike *quantitative*, species differences are uncommon. We observed human-specific spine clusters that—to the best of our knowledge—have not been described before. These clusters certainly deserve further histological and functional characterization that was beyond the scope of this

study. Spine clusters may be potentially fertile ground to explore new modes of synaptic and dendritic signaling. Some features—larger size and possibly more postsynaptic densities—represent quantitative expansion, but others may represent novel physiological phenomena. First, unlike branched or calyx spines (*Lee et al., 2005*; *Loschky et al., 2022*), several synaptic contacts would be exposed to membrane potentials and cytosolic plasticity factors in common. Second, while not quantified here, clustered inputs may operate in variable isolation from the dendrite and other local synapses as the neck diameter varies widely from the conserved value of 0.1–0.3 μm (*Harris and Stevens, 1988*; *Ofer et al., 2021*) to 0.5–0.7 μm (*Figure 2A, I*). The latter would have low electrical resistance, more contiguity with the dendritic shaft, and therefore behave like a small dendritic sub-branchlet. Indeed, spine clusters protrude ~1.5–3 μm, echoing the 2–4 μm length of the smallest terminal dendritic branchlets in the mouse. These two features may combine to produce both an associative structure, like proposed for branched spines (*Loschky et al., 2022*), and a new form of amplifying structure providing the substrate to initiate dendritic calcium spikes (*Roome and Kuhn, 2018*; *Eilers et al., 1995*; *Hartell, 1996*).

In addition to spine clusters, human PCs are more spine dense and thus may receive disproportionately more inputs than rodent PCs. The spine density we observe in mouse differs from studies reporting 1–2/μm—or the frequently cited work in rat reporting, by extrapolation, ~17/μm (*Napper and Harvey, 1988*)—but are in perfect agreement with recent confocal (*O'Brien and Unwin, 2006*) and EM reconstructions (*Nguyen et al., 2023*; *Loschky et al., 2022*; *Hillman and Chen, 1984*; *Ichikawa et al., 2016*; *Lu et al., 2009*; *Špaček and Hartmann, 1983*). Our data reveal elevated spine density in distal compartments, which comports with the knowledge that distal compartments receive denser PF innervation (*Eccles et al., 1967*). We also show that branched spine density increases distally, possibly supporting their proposed role in associative PF plasticity, a critical feature of the perceptron model (*Loschky et al., 2022*; *Brunel et al., 2004*). Larger spine heads in human may strengthen postsynaptic signaling so a higher percentage of inputs can influence PC output than in mouse. Alternatively, this amplification may simply compensate for human PC size, an idea supported by the fact that in human, unlike mouse, the spines are larger in distal compartments. Possibly reflecting shared factors of increased computational complexity and compensation for size or compartmentalization (*Beaulieu-Laroche et al., 2018*), pyramidal cells have similarly greater spine density (pyramidal: 1.3× vs Purkinje: 1.41×) and size (1.27× vs 1.41×) (*Benavides-Piccione et al., 2002*; *Benavides-Piccione et al., 2013*).

## Human Purkinje dendrite size and input numerosity

The previously reported average human PC dendritic length was 10–20,000 μm, varying with methodology (Golgi impregnation [*Louis et al., 2014*; *Paula-Barbosa et al., 1980*] vs post-fix dye electroporation [*Masoli et al., 2024*]). Our current knowledge, then, would conclude that human PCs are the same size as pyramidal cells (L2/3 are ~14,500 μm, <22,000 μm [*Mohan et al., 2015*]; thick tufted L5/6 are ~1–2000 μm longer [*Mohan et al., 2015*]; hippocampal CA1 are ~18,600 μm, <27,000 μm [*Mertens et al., 2024*]). While one might naturally expect the 2D radiations of PCs to make them larger in human than 1D radiating pyramidal cells, the previous data would suggest instead that there is a ceiling for maximal neuronal size and that, in human, both cell types met that limit before diverging. On the contrary, we demonstrate for the first time that PCs are substantially bigger than pyramidal cells and boast the largest dendritic arbor of any recorded neuron in the human brain. Intriguingly, our findings also indicate that pyramidal and PCs scale differently in mouse (~5500 vs ~6000 μm) than human (~15,000 vs ~65,000 μm).

With higher spine density and longer spiny dendrites, we can extrapolate that human PCs likely host half a million spines compared to ~30,000 in mouse. Branched spines (~7% of spines in both species) can host synapses with two PF axons while spine clusters (~10% in human) may host over four. As such, the human may receive ~750,000 synaptic inputs on spines. It is not only staggering to consider a single neuron receiving ~1 million inputs, but it highlights that the species' dendritic length ratio (11:1) belies an input ratio around 30:1.

## Human PCs multiplex CFs and other inputs

We previously found more multi-innervated PCs in posterior lobules of mouse (*Busch and Hansel, 2023*), likely due to the heightened prevalence of multi-branched PCs. This was recently replicated, albeit tangentially (*Nakayama et al., 2024*). We predicted that multi-branched structure permits

CF multi-innervation by providing segregated synaptic territories where multiple CFs could avoid the competitive pruning that would have prevented their translocation from soma to dendrite. This prediction was also supported by our finding that surplus CFs had greater synaptic strength in older animals (*Busch and Hansel, 2023*), possibly reflecting a delayed or elongated maturation process following distinct developmental trajectories of presynaptic plasticity at weak vs strong CF inputs before P21 (*Pätz et al., 2019*).

Human PC dendrites commonly ramify in more horizontal eccentricities than in mouse. This produces a distinct multi-branched motif with primary dendrites ramifying outward and giving rise to numerous (around 7–8) sub-compartments, each ~5–10,000 µm in length—roughly an entire rodent cell. Dendrite width complements multi-branched structure to further enable CF multi-innervation in mouse (*Busch and Hansel, 2023*). Here, co-immunolabeling of calbindin and peripherin (*Errante et al., 1998*) provides the first demonstration of CF multi-innervation in adult human. It also extends to humans the hypothesis that multi-branched structure supports CF competition avoidance. Finally, our co-labeling supports a likely higher prevalence of multi-innervation than in mouse.

Cerebellar cortical geometry allows us to speculate that a horizontally aligned dendritic arbor—often almost a millimeter wide (*Figure 1A and G*)—would allow human PCs to receive input from highly segregated excitatory PFs from distant granule cells. If conserved, spatial-functional PF clustering reported in mouse (*Wilms and Häusser, 2015*) would make their functional similarity diminish with distance. Thus, the disproportionate width of human PCs may sample more distinct PF representations. Similarly, this dendritic width may engender more compartment-specific inhibition by molecular layer interneurons (*Lackey et al., 2024*) and oversampling of modulatory influence by vertically ascending cholinergic beaded fibers (*Barmack et al., 1992*) and Bergmann glia (*De Zeeuw and Hoogland, 2015*). Taken together, human PCs likely perform highly complex input multiplexing and dendritic computations that are unlike what is commonly reported in mouse. This would, in turn, distinguish human PC dendritic plasticity and output information content as distinct from mouse PCs.

Studies of PC computation have largely considered the dendritic arbor a continuous functional compartment due to the equally unusual singularity and numerousness of CF and PF input, respectively. More recently, experimental and modeling studies demonstrate that Purkinje dendrite compartments exhibit heterogeneous ion channel density and plasticity (*Ohtsuki, 2020*; *Ohtsuki et al., 2012*), localized calcium signals (*Roome and Kuhn, 2018*; *Kitamura and Häusser, 2011*), PF input clustering (*Wilms and Häusser, 2015*), and variable calcium signaling due to branch shape in a homogeneous channel model (*Zang et al., 2018*; *Zang and De Schutter, 2021*; *Cirtala and De Schutter, 2024*). Combined with the finding that some PC dendrites are innervated by multiple CFs that confer branch-specific signaling (*Busch and Hansel, 2023*), the potential for dendritic input multiplexing is becoming more widely appreciated (*Zang and De Schutter, 2021*).

We previously suggested that CF multi-innervation may be an adaptation to maintain receptive field matching in multi-branched PCs with more disparate PF representations across segregated dendritic compartments (*Busch and Hansel, 2023*). This would preserve perceptron function among mouse PCs with expanded dendritic arbors. Our findings extend this concept to human PCs where there is likely a greater demand to match distinct receptive fields. As a possible consequence, human PC output may convey more features of sensory, internal, or cognitive state. Individual CFs convey a variety of signals (*Gaffield et al., 2019*; *Ju et al., 2019*; *Kostadinov et al., 2019*). PF-dependent simple spike modulation reflects unidimensional movement kinematics in oculomotor L6–7 (*Pi et al., 2024*), but can represent multiple proprioceptive features in anterior vermis during passive vestibular stimulation (*Zobeiri and Cullen, 2022*) and reinforcement learning error and learning state during a visuomotor association task among Crus1-2 PCs (*Sendhilnathan et al., 2022*) where multi-branched structure is more common. Thus, multi-branched PC computation may resemble mixed selectivity encoding by pyramidal cells involved in complex decision-making (*Tye et al., 2024*). Importantly, expansion of input numerosity or strength does not necessarily improve computational capacity. Excess CF innervation is linked to cellular and behavioral sensory over-responsiveness in mouse autism models (*Simmons et al., 2022*; *Busch et al., 2023*). Human essential tremor is characterized in part by excess CF innervation from ectopic lateral crossings (*Wu et al., 2021*) and translocation to thin dendrites (*Pan et al., 2020*).

## Human cerebellum may harness regional PC demographics and subregional clustering to generate task-specific computation

We previously described a parasagittal gradient of PC morphological demographics in human and mouse mid-hemisphere (*Busch and Hansel, 2023*). Here, we control for a possible role of parasagittal developmental gradients (*Beekhof et al., 2021*) both through comparison with the vermis—with a distinct demographic gradient despite shared parasagittal development—and by showing that inter-hemisphere demographic similarity covaries with functional symmetry (*Nettekoven et al., 2024*; *Buckner et al., 2011*; *Stoodley and Schmahmann, 2009*; *Wang et al., 2013*; *Wang et al., 2014*; *Krienen and Buckner, 2009*). In mouse, posterior hemisphere processes multisensory information from mossy fibers (*Huang et al., 2013*; *Ishikawa et al., 2015*; *Shimuta et al., 2020*) and CFs (*Simmons et al., 2022*; *Busch et al., 2023*; *Gaffield et al., 2019*; *Ju et al., 2019*; *Bosman et al., 2010*; *Ohmae and Medina, 2015*; *Rondi-Reig et al., 2014*; *De Zeeuw et al., 1998*) and receives stronger somatosensory feedback from neocortex than the vermis (*Pisano et al., 2021*). In human, lateral and posterior regions are more responsive to cognitive, affective, and sociolinguistic information (*Stoodley and Schmahmann, 2009*; *LeBel and D'Mello, 2023*; *Van Overwalle et al., 2015*; *LeBel et al., 2021*) and, intriguingly, exhibit stronger hyper-allometric expansion than the cerebrum and the rest of cerebellum in primates (*Magielse et al., 2023*).

We propose that the relative proportions of multi-dendritic structure and horizontal orientation act as complementary toggles to generate four classes of morphological demography. Surprisingly, the regional borders of each class align well with four functional classes from a new consensus atlas of human cerebellum (*Nettekoven et al., 2024*; *Figure 6—figure supplement 2*). Furthermore, regions with more complex morphology (increased multi-branched and horizontal orientation) roughly match regions processing information that is more cognitive (co-active with non-primary sensory neocortical regions) and associative (requiring integration and association of multi-modal inputs) in nature (*Figure 6—figure supplement 2*).

Subregional clustering is more prevalent in human and only partly reflective of developmental tissue foliation. The spatial scale of clustering and the comparative scale ratio (~5–7× longer in human; enhanced clustering in core regions of 2 mm vs 400 µm) are congruent with the absolute and relative widths of the dendritic arbor. Both properties support the idea that neighboring cells' primary dendrites may reciprocally influence each other during development. Notably, the clustering scale ratio exceeds the comparative PC height ratio (a proxy for allometric constraint). This comports with observations that segregated branches of multi-dendritic mouse PCs often stagger in interdigitated dendrite arbors with lower medio-lateral overlap (*Nedelescu et al., 2018*). This suggests that cerebellar cortical region, especially in human, may employ 'patches' of homogeneous PC morphologies to create niches for heterogeneous computations. Nonrandom spatial clustering of neuronal cells by morphological subtype has not previously been described in other brain areas, potentially making this a cerebellum-specific means to boost functional diversity.

## Methods

### Subjects

Human cerebellar tissue was collected from three unembalmed and three embalmed donor bodies provided by the Anatomical Gift Association of Illinois (AGAI) and the New York Brain Bank (NYBB). Unembalmed individuals were 93 (F), 61 (M), and 37 (M) and died of causes unrelated to cerebellar morphology (e.g. chronic obstructive pulmonary disease, bile duct cancer, and hypertrophic cardiomyopathy, respectively). The first two cadavers were stored for 6 and 7 days before fixation. Embalmed individuals were 92 (F), 95 (F), and 86 (M) years old and died of causes unrelated to cerebellar morphology (e.g. 'failure to thrive', likely 'failure to thrive', and colon cancer, respectively). Cadavers were stored for 2, 6, and 2 months, respectively. During life, all study subjects signed an informed consent approved by the AGAI or NYBB.

For experiments involving mice, all experimental and surgical procedures were in accordance with the University of Chicago Animal Care and Use Committee guidelines. We used wildtype C57BL/6J mice housed on a 12 hr light/dark cycle. Animals of either sex were used in all experiments and no sex-dependent differences were observed in any reported measures.

## Human tissue preparation

One hemisphere from each of the embalmed specimens was used previously (*Busch and Hansel, 2023*). Preparation of the vermis and opposing hemisphere from the embalmed specimens and of one hemisphere from the unembalmed specimens followed the same procedure as previously reported. In brief, whole cerebellums were fixed in 4% paraformaldehyde (PFA) for 1 week after being obtained, regardless of whether they were embalmed or unembalmed. Specimens were sectioned by hand in the parasagittal axis to obtain 3–5 mm blocks that were sometimes cut transversely and further fixed for 2–4 days. Blocks were rinsed in 0.01 M phosphate-buffered saline (PBS) and sliced at 50 μm in the parasagittal plane with a vibratome (Leica VT-1000S). Slices selected for immunolabeling were photobleached at 4°C for 3–4 days to reduce autofluorescence.

## Mouse tissue preparation

To obtain sparse PC labeling for single-cell dendrite and spine reconstruction in mice, we used tissue from animals expressing PC-specific GCaMP6f label for unrelated imaging studies. Surgeries were performed as described previously (*Busch and Hansel, 2023*) on animals aged 10–12 weeks under ketamine/xylazine anesthesia (100 and 10 mg/kg) with subcutaneous injections of meloxicam (1–2 mg/kg), buprenorphine (0.1 mg/kg), and sterile saline (0.5–1 mL) as above. Body temperature was maintained at 35–37°C with a feedback dependent heating pad. The skin above the posterior skull was excised and the bone cleaned to implant a metal headframe over the interparietal bone via dental cement. After 3–4 days of recovery, mice were anesthetized and a 4 mm craniotomy and durectomy was made at 2.5 mm lateral from midline and 2.5 mm caudal from lambda, exposing cerebellar simplex, Crus I, and anterior Crus II. A glass microelectrode with ~300 μm tip diameter was used to inject a viral cocktail with low titer PC-specific L7-Cre (0.5%, AAV1.sL7.Cre.HA.WPRE.hGH.pA; Princeton Neuroscience Institute [PNI] Viral Core Facility; acquired from the lab of Dr. Samuel Wang, Princeton University) and high titer Cre-dependent GCaMP6f (20%, AAV.CAG.Flex.GCaMP6f.WPRE. SV40; Addgene, #100835) at a depth of ~300 μm below the pial surface of lateral Crus I (~900 nL per site, 5 min wait before needle retraction). A two-layer cranial window (4 mm inner window, Tower Optical; 5 mm outer window, Warner Instruments) was implanted over the craniotomy and sealed with dental cement (Metabond).

Imaging experiments were performed 2–3 weeks later, after which mice were anesthetized with ketamine/xylazine (100 and 10 mg/kg) and perfused with 4% PFA. Sleep/wake periods may influence spine densities (*Loschky et al., 2022*), but we controlled for this in part by collecting mouse tissue during their sleep cycle, which may compare well with our human cases that passed away either during sleep or a period of decreased metabolism. Cerebellums were removed and incubated for 2 hr in 4% PFA at 4°C and then overnight in 30% sucrose in 0.1 M PB at 4°C (until the tissue sank from the surface). The tissue was then rinsed briefly in 0.1 M PB, dried and blocked, submerged in OCT medium, flash-frozen, and then sliced (50 μm, parasagittal plane) using a cryostat microtome (CM 3050S, Leica). Sparseness was not required to quantify cell morphology demographics across the vermis, so tissue was obtained from wildtype C57BL/6J mice without previous surgery and were anesthetized, perfused, and sliced as above.

## Calbindin immunohistochemistry

Either unembalmed (for reconstructions) or embalmed (for regional analysis) tissue was washed in 50 mM glycine in 0.01 M PBS for 2 hr at 4°C and incubated in 20 mM sodium citrate in 0.01 M PBS at 50–60°C using a heated water bath for 30 min. After cooling to room temperature (RT), tissue was washed in 20 mM sodium citrate for 5 min then rinsed 2×30 s in dH$_2$O. Next, slices were permeabilized at RT in 0.01 M PBS containing 0.025% Triton-X (PBS-TX) for 1 hr. Blocking was done with PBS-TX containing 5% normal donkey serum (NDS) and 5% bovine serum albumin (BSA) for 1 hr at RT followed by incubation in guinea pig anti-calbindin primary antibody (1:1000; Synaptic Systems Cat# 214 004, RRID:AB_10550535) solution overnight (18–20 hr) at 4°C with 1% NDS in PBS-TX. After 3×10 min washes in PBS-TX at RT, slices were incubated in donkey anti-guinea pig Cy3 secondary antibody (1:200; Jackson ImmunoResearch Labs Cat# 706-165-148, RRID:AB_2340460) for 2 hr at 4°C with 1% NDS in PBS-TX. Finally, slices were washed in PBS-TX for 3×10 min, mounted and coverslipped with Vectashield (Vector Laboratories, Inc), and allowed to set overnight before visualization.

For calbindin-based labeling of PCs in mouse tissue without GCaMP6f, the same procedure was used as above with some small changes: glycine incubation for 1 hr instead of 2 hr and heated sodium citrate incubation for 20 min instead of 30 min. Slices were incubated in primary antibody solution with guinea pig anti-calbindin (1:1000), then in secondary antibodies with donkey anti-guinea pig Cy3 (both 1:200).

## Dual calbindin and peripherin immunohistochemistry

VGluT2 and cocaine- and amphetamine-regulated transcript (CART) (*Press and Wall, 2008*; *Reeber and Sillitoe, 2011*) label CFs in human (*Lin et al., 2014*; *Wu et al., 2021*), but are restricted to terminal boutons and minor processes in the molecular layer, precluding disambiguation of mono- and multi-innervation without an olivary tracer (*Miyazaki and Watanabe, 2011*; *Busch and Hansel, 2023*). Some reports used Golgi impregnation to visualize human CFs (*Ramón y Cajal, 1909*; *Marin-Padilla, 1985*), but this technique cannot visualize CF-PC pairs reliably beyond the molecular layer in postnatal tissue due to myelination. To distinguish fiber sources, we instead labeled CF axonal fibers with the intermediate neurofilament protein peripherin (*Errante et al., 1998*). Unembalmed human tissue was washed in 200 mM glycine in 0.01 M PBS for 2 hr at RT and incubated in 10 mM sodium citrate in 0.01 M PBS at 80–90°C using a heated water bath for 30 min. After cooling to RT, tissue was washed in 0.5% Tween-20 in 0.01 M PBS (PBS-Tween) for 3×5 min. Next, slices were permeabilized at RT in 0.01 M PBS containing 2.5% Triton-X (PBS-TX) for 1 hr and then incubated in 200 mM glycine in PBS-Tween for 15 min. Blocking was done with PBS-Tween containing 10% NDS and 5% BSA for 2 hr at RT followed by incubation in polyclonal guinea pig anti-calbindin (1:500; Synaptic Systems Cat# 214 004, RRID:AB_10550535) and polyclonal rabbit anti-peripherin (1:500; EnCor Biotechnology Cat# RCPA-Peri, RRID:AB_2572375) primary antibody solution overnight (18–20 hr) at 4°C and then at RT for 3–4 hr with 1% NDS in PBS-Tween. After 3×10 min washes in PBS-Tween at RT, slices were incubated in donkey anti-guinea pig AF488 (1:200; Jackson ImmunoResearch Labs Cat# 706-545-148, RRID:AB_2340472) and donkey anti-rabbit Cy3 (1:200; Jackson ImmunoResearch Labs Cat# 706-165-152, RRID:AB_2307443) secondary antibody solution for 2 hr at RT with 1% NDS in PBS-Tween. Finally, slices were washed in PBS-Tween for 3×10 min, mounted and coverslipped with Vectashield (Vector Laboratories, Inc) and allowed to set overnight before visualization.

## Confocal imaging for cell, spine, and peripherin fiber reconstruction

Following immunolabeling of unembalmed human tissue and perfusion of GCaMP6f labeled mouse tissue, we selected PCs within L6–8 of the mid-hemisphere for their lack of truncation of their dendritic arbor and isolation from adjacent cells to minimize the chance of misattributing a branch from another cell. Cells and spines were then manually reconstructed using NeuronStudio (*Rodriguez et al., 2003*). No shrinkage factor or z-correction was applied.

### Cell reconstructions

We collected z-stack tile scans of individual PCs at ×40 (Zeiss EC Plan-Neofluar 1.3NA, oil immersion) with a confocal microscope (Zeiss LSM 900, Examiner.Z1) with 1× digital zoom, 2× line averaging, producing a 0.2079 µm × 0.2079 µm × 0.5 µm voxel resolution.

### Cell and peripherin fiber reconstruction

Multi-channel z-stack images of PCs and peripherin fibers were collected at ×40 (Zeiss EC Plan-Neofluar 1.3NA, oil immersion; Zeiss LSM 900, Examiner.Z1) with 0.72× digital zoom, 2× line averaging, and 0.289 µm × 0.289 µm × 1 µm voxel resolution.

### Spine reconstruction

Single z-stack images of branch segments from distal, intermediate, and proximal compartments for spine reconstruction were collected at ×63 (Leica HC PL APO 1.4 UV, oil immersion; Leica Stellaris 8 laser scanning confocal microscope; University of Chicago Integrated Light Microscopy Core RRID:SCR_019197) with 7× digital zoom, 4× line averaging, and 25.76 nm × 25.76 nm × 0.15 µm voxel resolution. As some spine measurements presented here approach the limits of our confocal resolution, we identify the diffraction limit of our imaging method via a calculation of the Rayleigh criterion (modestly more conservative than Abbe's limit with a 0.5 constant) for the minimum resolvable

distance (R) given the fluorophore excitation wavelength ($\lambda$; 555 nm for the Cy3 probe) and numerical aperture of the objective (NA) as follows:

$$R = (0.61 \times \lambda)/NA$$

$$R = (0.61 \times 555)/1.4 = 241.8 \, nm$$

Branch eccentricity was calculated by translating the origin of every branch to (0,0) in the coordinate plane and taking the slope of the best fit line for all branch trace vertices. The slope was mirrored over the x-axis for branches projecting downward toward the PC layer (in quadrants 3 and 4 of the coordinate plane). As it made no difference whether the branch projected to the left or right, all branches projecting leftward (in the second quadrant of the coordinate plane) were mirrored over the y-axis for simplicity. Thus, the eccentricity of every branch relative to the horizontal plane of the PC layer was maintained while the direction was reversed.

Each instance of an intact co-labeled PC and peripherin fiber allowed us to make one of four observations: 'putative mono-innervation', in which a single fiber approaches the target PC and branches to run in apposition to all major primary dendrites (*Figure 3A*); 'absence multi-innervation', in which a single fiber approaches and runs in apposition with some primary dendrites but is conspicuously absent from others (*Figure 3B*); 'putative multi-innervation', in which the primary dendritic branches receive multiple unconnected fibers that are truncated so we could not observe their independence in the GCL (*Figure 3C*); and 'fully labeled multi-innervation', in which multiple labeled fibers approach the PC from the GCL and travel to distinct primary dendrites either entirely separately, or following a brief distance of shared apposition to a primary dendrite before diverging to different dendrites (*Figure 3D and E*).

## Slice reconstruction and cell counting

Parasagittal slices were traced and cells were mapped as described previously (*Busch and Hansel, 2023*). Briefly, slides were visualized under ×10 or ×20 magnification (Zeiss Achroplan 0.25NA, air; Olympus UMPlanFL N 0.5NA, water) and illuminated with an epi-fluorescent light source (LEJ HBO-100). We manually scanned through the cerebellar cortex and classified PCs by their dendritic morphology and their location by foliar subregion (e.g. gyrus, bank, and sulcus), both based on criteria listed below. To mark the morphology and cell location accurately in both human and mouse tissue, we initially traced the outlines of the pial surface, white matter tracts, and PC layer of the whole parasagittal section. Cells were only included for categorization if the soma and at least 200 μm lengths of primary dendritic trunks were clearly labeled such that all features of Normative, Split, and Poly and Vertical or Horizontal categories were unambiguously present or absent (see criteria below). We marked the location and morphological type of each cell in the slice map and scanned this map as an input image to a custom MATLAB GUI where each point's X,Y coordinate, foliar location, and morphological category could be digitized. The output data were imported to R for downstream analysis and plotting.

## PC morphological category definitions and criteria

The criteria for morphology category definitions were the same as previously (*Busch and Hansel, 2023*) but we reiterate the full description here for clarity. In human, PCs were deemed Normative if they had the following features: (1) a single trunk emerging from the soma, and (2) either no bifurcation of the primary trunk within two soma distances (2× the diameter of the soma, 25–35 μm per soma) or a highly asymmetrical bifurcation where the smaller branch did not project in the parasagittal axis more than 200 μm from the main dendritic compartment. PCs were defined as Split if they had the following features: (1) a single trunk emerging from the soma, and (2) either symmetrical bifurcation of the primary trunk within two soma distances or an asymmetrical bifurcation within two soma distances where the smaller branch projected more than 200 μm from the main dendritic compartment and thus reached prominence by its overall length and sub-branching. PCs were defined as Poly if they had more than one trunk emerging from the soma regardless of relative size.

In mouse, PC categories were defined the same way, except that the bifurcation threshold of two soma distances (each soma diameter is 18–22 μm) was set at 40 μm, and the smaller branch of an asymmetrical bifurcation had to project only 100 μm away from the main dendritic compartment.

In mouse and human, Split and Poly PCs were further subdivided into Vertical or Horizontal ramification patterns. Split and Poly PCs were defined as Horizontal if one of two primary dendrites ramified parallel with the PC layer for >300 µm in human (>150 µm in mouse), or both primary dendrites ramified in opposing directions parallel with the PC layer for >150 µm each in human (>75 µm in mouse). Dendrites were considered parallel if the dendrite, at 300 or 150 µm from the soma respectively, ramified at <30° from the top of the PC layer. Otherwise, the cell was defined as Vertical.

## Foliar subregion category definitions and criteria

The criteria for foliar category definitions were the same as previously (*Busch and Hansel, 2023*) but we reiterate the full description here for clarity. PC locations were defined as either Gyrus, Bank, or Sulcus based on the relative expansion/compression of the granule cell/molecular layers in the parasagittal axis. Gyrus was defined as a region where the total parasagittal length of the pial surface exceeded that of the border between the granule cell layer and the white matter. Bank was defined as regions where those two lengths were equal, such that neither layer of the cortex was compressed or expanded relative to the other. Finally, Sulcus was defined as regions where the total parasagittal length of the pial surface was less than that of the border between the granule cell layer and the white matter. Both intermediate sulci, embedded within a continuous Bank region, and full sulci were combined for these analyses.

## Spine morphology criteria and volume calculations

Manual reconstruction of spines was performed using NeuronStudio and by scanning through a z-stack image of a spiny branch to better visualize the relations between contiguous and sometimes noncontiguous structures surrounding the dendrite. Spines were categorized as either thin or mushroom spines if they had a classical head and neck structure with a head that had either a less than or greater than 500 nm diameter, respectively. Spines were categorized as branched if two heads emerged from a shared neck emerging from the dendrite. Spines were categorized as 'spine clusters' if they had a single head with discontinuous and/or bumpy structure that produced three or more puncta with distinct prominences from the core head matrix.

Spine volume as a fraction of volume surrounding the dendrite was calculated by taking the sum of the spine head volumes for each branch segment and dividing it by the volume of the surrounding cylindrical space, which in turn was the subtraction of the cylindrical volume of the dendrite (using the mean dendritic radius) from the larger cylindrical volume with a radius combining the mean dendrite radius and the mean spine head protrusion distance.

## Cell morphology clustering analysis

The output dataset from slice reconstruction and cell counting contained cell ID information paired with X,Y coordinates in the slice. To calculate cell-type clustering, we wrote a custom R script to measure cluster scores for each cell based on either immediately adjacent cells or cell populations.

### Adjacency clustering

First, for each dataset (one slice each from vermis and the mid-hemisphere for each individual), the Euclidean distance was calculated between each consecutive pair of cells along the PC monolayer from anterior to posterior. Second, an initial cluster score of –1 was assigned to each cell. Third, the final cluster score was assessed for each cell based on whether the leading or following immediately adjacent cell matched the morphology of that cell. If there were adjacent cells within the threshold distance (1000 µm for human, 200 µm for mouse), then the cluster score was set at 0. Leading or following adjacent cells with matching morphologies each added (+1) to the cluster score, producing either a final score of 1 if one of two adjacent cells match or 2 if both are matching. The morphology match was determined one of several different ways, requiring either a complete match (e.g. both the morphological type of Normative, Split, or Poly and the orientation of Vertical or Horizontal are the same; thus, cells fall into five total categories of Normative, vertical Split, horizontal Split, vertical Poly, and horizontal Poly, wherein a vertical Split cell does not match with a horizontal Split cell or a vertical Poly cell), or a more liberal match by only morphological type (e.g. Normative, Split, or Poly; thus, a vertical Split cell matches with a horizontal Split cell but not a vertical Poly cell) or orientation (e.g. vertical or horizontal; thus, a vertical Split cell does not match with a horizontal Split cell but

does match a vertical Poly cell). Once cluster scores were calculated, consecutive IDs were assigned to each cluster such that the number of cells and total parasagittal inter-somatic distance could be determined. The identical operation was performed for a shuffled version of each dataset to compare the observed effect of clustering with chance based on cell-type ratios. All aspects of cell information were held constant (e.g. X,Y coordinate, location by foliar subdivision) and the morphology was shuffled without replacement (i.e. the same ratio of cell types was maintained). To assess the specific effect of foliar location on clustering, the same analysis was performed but using a shuffled dataset in which the possible shuffled cell-type identities for each cell was only drawn from the cells within the same foliar subdivision (e.g. gyrus, bank, or sulcus).

## Population clustering

Here, the number of cells of every type was tallied among the whole population of cells within a defined distance of each cell in question. This threshold varied from 250 μm and 50 μm in human and mouse, respectively, to 10 mm and 2 mm, respectively. Thus, this analysis is more lenient to interruptions in an otherwise relatively homogeneous population by assessing multiple cells and not just the single most adjacent leading and following cells. The frequency of observing each cell type in the population around each cell type was determined. From that was subtracted the same calculation for a version of each dataset that was shuffled as described above without considering foliar location (e.g. new shuffled identities were drawn from the whole population, not just from those cells in the same foliar subdivision). This subtraction gave us a percent relative increase in the rate of observing either the one complete matching morphology, or any of all four nonmatching morphologies, vs the rate expected by chance. Then, across all cell types, we averaged the rates of all matches (e.g. combining the rate of vertical Splits near a vertical Split with the rate of horizontal Polys near horizontal Polys, etc.) and non-matches (e.g. combining the rate of vertical Splits near a horizontal Polys with the rate of vertical Polys near Normative, etc.).

When measuring wider populations with larger radius thresholds, we controlled for the effect of local clustering on the rate of clustering among distant cells by instead assessing cell populations selectively in 500 μm leading and following shell regions around each cell while ignoring the most immediately local cells within the core region (from the soma location to the inner edge of the shell analysis region) around the cell in question. For example, to analyze clustering in a 500–1000 μm shell region, we only included cells that were at least 500 μm away but no more than 1000 μm away on either side (leading and following the cell in question), and thus ignored more local cells within a core 1000 μm (500 μm on either side) of the cell in question. Similarly, to analyze clustering in a 1500–2000 μm shell region, we only included cells that were at least 1500 μm away but no more than 2000 μm away on either side (leading and following), and thus ignored more local cells within a 3000 μm core region around the cell. By calculating the elevation of clustering over chance for equivalently sized shell regions with variable distances running stepwise in 500 μm increments while ignoring a growing core region, we could selectively isolate distant populations and observe the true drop-off distance for clustering.

## Statistics

Standard parametric statistics such as the Student's t-test or ANOVA with Tukey post hoc and Bonferroni correction were used to assess individual and multi-group comparisons except in cases where the data were non-normally distributed, in which case we used single or multiple comparisons Mann-Whitney U tests. In cases of a paired comparison where the underlying inter-individual variability was uninformative, we performed a one-way Student's t-test on within-individual normalized data. A Kolmogorov-Smirnov test was used to assess differences between cumulative distributions. A chi-squared test for independence was used to distinguish the ratios of categorical data by group. Co-variation of each measure by sex was assessed but no significant differences were observed.

## Acknowledgements

For valuable advice and technical support, we thank Hansel lab members T-F Lin, A Silbaugh, D Huang, and A Ferrell. We thank RA Eatock, W Wei, M Sheffield, and P Mason (UChicago Neurobiology) for insightful discussions and crucial feedback on the manuscript. Human tissue was made available by the Anatomical Gift Association of Illinois and by the New York Brain Bank with the assistance of Dr.

Phyllis Faust (Columbia University). This work was supported by National Institutes of Health (NINDS) grant R21NS124217 (CH) and F31NS129256 (SEB).

## Additional information

### Funding

| Funder | Grant reference number | Author |
|---|---|---|
| National Institute of Neurological Disorders and Stroke | R21NS124217 | Christian Hansel |
| National Institute of Neurological Disorders and Stroke | F31NS129256 | Silas E Busch |

The funders had no role in study design, data collection and interpretation, or the decision to submit the work for publication.

### Author contributions

Silas E Busch, Conceptualization, Data curation, Formal analysis, Funding acquisition, Investigation, Visualization, Methodology, Writing - original draft, Writing – review and editing; Christian Hansel, Conceptualization, Supervision, Funding acquisition, Project administration, Writing – review and editing

### Author ORCIDs

Silas E Busch ⬧ https://orcid.org/0000-0001-6720-1921
Christian Hansel ⬧ https://orcid.org/0000-0001-5750-7097

### Ethics

For experiments involving mice, all experimental and surgical procedures were in accordance with the University of Chicago Animal Care and Use Committee guidelines.

Reviewer #1 (Public review): https://doi.org/10.7554/eLife.105013.3.sa1
Reviewer #2 (Public review): https://doi.org/10.7554/eLife.105013.3.sa2
Author response https://doi.org/10.7554/eLife.105013.3.sa3

## Additional files

### Supplementary files

MDAR checklist

### Data availability

All data generated and analyzed during this study, along with code, is included in the manuscript and available in a Zenodo repository: https://doi.org/10.5281/zenodo.15066697.

The following dataset was generated:

| Author(s) | Year | Dataset title | Dataset URL | Database and Identifier |
|---|---|---|---|---|
| Busch S | 2025 | Non-allometric expansion and enhanced compartmentalization of Purkinje cell dendrites in the human cerebellum | https://doi.org/10.5281/zenodo.15066697 | Zenodo, 10.5281/zenodo.15066697 |

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
