## [Editor Report · eLife Assessment]

This is a **convincing** study of the morphological properties of Purkinje cell dendrites and dendritic spines in adult humans and mice, and the anatomical determinants of multi-innervation by climbing fibers. The data will provide an **important** resource for the field of cerebellar computation.

---

## [Referee Report · Reviewer #1 (Public review)]

Summary:

Busch and Hansel present a morphological and histological comparison between mouse and human Purkinje cells (PCs) in the cerebellum. The study reveals species-specific differences that have not previoulsy been reported despite numerous observations in these species. While mouse PCs show morphological heterogeneity and occasional multi-innervation by climbing fibers (CFs), human PCs exhibit a widespread, multi-dendritic structure that exceeds expectations based on allometric scaling. Specifically, human PCs are significantly larger, exhibit increased spine density, with a unique cluster-like morphology not found in mice.

Strengths:

The manuscript provides an exceptionally detailed analysis of PC morphology across species, surpassing any prior publication. Major strengths include a systematic and thorough methodology, rigorous data analysis, and clear presentation of results. This work is likely to become the go-to resource for quantitation in this field. The authors have largely achieved their aims, with the results effectively supporting their conclusions.

Weaknesses:

There are a few concerns that need to be addressed, specifically related to details of the methodolology as well as data interpretation based on the limits of some experimental approaches. Overall, these weaknesses are minor.

Comments on revisions:

The authors addressed my concerns in the revised manuscript. One bit of clarification, the defraction limit calculation involves the wavelength of light used for excitation not emission ("...for the minimum resolvable distance (R) given the fluorophore emission wavelength [l; 570nm for the Cy3 probe] and numerical aperture of the objective (NA) as follows:"). This is why a 2p system has less resolving power than a confocal system as it uses much longer wavelengths for excitation.

---

## [Referee Report · Reviewer #2 (Public review)]

Summary:

This manuscript follows up on a previously published paper (Busch and Hansel 2023) which proposed that the morphological variation of dendritic bifurcation in Purkinje cells in mouse and human is indicative of the number of climbing fiber inputs, with dendritic bifurcation at the level of the soma resulting in a proportion of these neurons being multi-innervated. The functional and anatomical climbing fiber data was obtained solely from mice, since all human tissue was embalmed and fixed, and the extension of these findings to human Purkinje cells was indirect. The current comparative anatomy study aims to resolve this question in human tissue more directly and to further analyse in detail the properties of adult human Purkinje cell dendritic morphology.

Strengths:

The authors have carried out a meticulous anatomical quantification of human Purkinje cell dendrites, in tissue preparations with better signal to noise ratio than their previous study, comparing them with those from mice. They show that human PC dendrites are much larger than would be expected from straightforward scaling to brain size and, importantly, they now present immunolabelling results that trace climbing fiber axons innervating human PCs in a subset of the data. As well as providing detailed analyses of spine properties and interesting and unexpected new findings of human PC dendritic length and spine types, the work suggests that human PCs that have two clearly distinct dendritic branches have an approximately 80% chance of receiving more than one CF input, segregated across the two branches. Albeit entirely observational, the data will be of widespread interest to the cerebellar field, in particular those building computational models of Purkinje cells.

Weaknesses:

The work is, by necessity, purely anatomical. It remains to be seen whether there are any functional differences in ion channel expression or functional mapping of granule inputs to human PCs compared with the mouse that might mitigate the major differences in electronic properties suggested.

Comments on revisions:

I am happy with the updated manuscript in response to my suggestions and I have no further comments.

---

## [Author Response]

The following is the authors’ response to the original reviews:

**Reviewer #1 (Public review):**
Summary:Busch and Hansel present a morphological and histological comparison between mouse and human Purkinje cells (PCs) in the cerebellum. The study reveals species- specific differences that have not previously been reported despite numerous observations of these species. While mouse PCs show morphological heterogeneity and occasional multi-innervation by climbing fibers (CFs), human PCs exhibit a widespread, multi-dendritic structure that exceeds expectations based on allometric scaling. Specifically, human PCs are significantly larger, and exhibit increased spine density, with a unique cluster-like morphology not found in mice.Strengths:The manuscript provides an exceptionally detailed analysis of PC morphology across species, surpassing any prior publication. Major strengths include a systematic and thorough methodology, rigorous data analysis, and clear presentation of results. This work is likely to become the go-to resource for quantitation in this field. The authors have largely achieved their aims, with the results effectively supporting their conclusions.

We are grateful to this reviewer for their thoughtful assessment that this work will be a go-to resource for the field.

Weaknesses:There are a few concerns that need to be addressed, specifically related to details of the methodology as well as data interpretation based on the limits of some experimental approaches. Overall, these weaknesses are minor.

We thank this reviewer for their careful reading of the manuscript and for highlighting limitations and weaknesses in the methodology. We are in full agreement that while interpretation is somewhat limited, there is still value in their description. As detailed below in response to this reviewer’s recommendations, we provide more description of our imaging resolution. This additional detail clarifies that our quantitation is appropriate for the scale of the objects being measured and provides critical information to help readers assess the findings as they may pertain to their own work.

**Reviewer #2 (Public review):**
Summary:This manuscript aims to follow up on a previously published paper (Busch and Hansel 2023) which proposed that the morphological variation of dendritic bifurcation in Purkinje cells in mice and humans is indicative of the number of climbing fiber inputs, with dendritic bifurcation at the level of the soma resulting in a proportion of these neurons being multi-innervated. The functional and anatomical climbing fiber data was obtained solely from mice since all human tissue was embalmed and fixed, and the extension of these findings to human Purkinje cells was indirect. The current comparative anatomy study aims to resolve this question in human tissue more directly and to further analyse in detail the properties of adult human Purkinje cell dendritic morphology.Strengths:The authors have carried out a meticulous anatomical quantification of human Purkinje cell dendrites, in tissue preparations with a better signal-to-noise ratio than their previous study, comparing them with those from mice. Importantly, they now present immunolabelling results that trace climbing fiber axons innervating human PCs. As well as providing detailed analyses of spine properties and interesting new findings of human PC dendritic length and spine types, the work confirms that human PCs that have two clearly distinct dendritic branches have an approximately x% chance of receiving more than one CF input, segregated across the two branches. Albeit entirely observational, the data will be of widespread interest to the cerebellar field, in particular, those building computational models of Purkinje cells.

We thank this reviewer for their positive and considered assessment of our work. We enthusiastically agree that while these data are descriptive in nature, they may be of interest across modalities of cerebellar research and will provide a more detailed framework for cross-species comparisons and single cell computational modeling, which remains a critical tool to explore the human case given the inaccessibility of physiological experimentation.

Weaknesses:The work is, by necessity, purely anatomical. It remains to be seen whether there are any functional differences in ion channel expression or functional mapping of granule inputs to human PCs compared with the mouse that might mitigate the major differences in electronic properties suggested.

We are in full agreement with the reviewer that the focused anatomical description of this manuscript could not make strong assertions about function given that cellular and circuit physiology is determined by many additional factors that remain unexamined. We appreciate that the reviewer acknowledges that this is out of necessity as those factors are inaccessible to experimentation at the current time; however, we are enthusiastic that our current findings will motivate future work that will shed light on these critical additional features of the system, both in rodents and humans.

**Reviewer 1 (Recommendations for the authors):**
PCs are now known to be genetically diverse, with unique PC types found only in humans. Could this cellular diversity contribute to the differences observed between species in this study? This possibility should be at least discussed in the context of the findings.

We agree that this is a fascinating possibility. The perhaps most detailed recent study (Sepp et al., Nature 625, 2024) – in a conservative assessment – describes four developmental PC subtypes in mice that are identical in humans. The study points out that the subtype ratio changes over the course of development, though. Taken together with the possibility of additional human-specific subtypes, a genetic basis for morphological as well as physiological diversity arises. This is now discussed on p. 7. It needs to be kept in mind, however, that other factors, such as push-pull influences during tissue growth, might also play a role.

The human tissue used in this study was obtained from elderly individuals, while the mouse tissue was not. It is unclear whether the age difference might influence the findings, and this warrants further discussion or control.

We share this concern, in particular regarding the spine / spine cluster analysis as here tissue quality and or degenerative effects might play a role. We additionally analyzed a tissue sample from a 37 year-old human, and observed the same spine clusters as in the other human brains. This is now described on p. 4 of the revised manuscript.

The study includes spine size comparisons, but it is not clear if the point spread function (PSF) of the microscope provides the necessary resolution for these quantitative assessments. For instance, are multi-headed spines truly multi-headed, or could this be an artifact of limited resolution?

This is an important point. We addressed it by calculating the Rayleigh limit (more conservative than the Abbe limit) as 248.4nm for the equipment and conditions used (Methods, p. 22). On pages 3-5, we updated our Results section accordingly to point out what quantifications are well supported and discuss the limitations (p. 3-5).

**Reviewer 2 (Recommendations for the authors):**
This is nice work which must have been very time-consuming. It would be good to make sure that the technical details are properly discussed, to quantify the data properly. Please include details of how you measured the resolution of the microscope used to evaluate spine size.

See our response to the last comment of Referee 1 above.

The figure panels are mostly satisfactory, but they are exceptionally crowded and will probably be difficult to read at the final size. Some work tidying these would be worth it. In Figure 3B, include mention of open and blue triangles in legend. In 3E, the dendritic branches are shown at a different gray scale. You have not done this elsewhere, so probably good to mention it in the legend.

Figure 3 and its legend have been updated / improved accordingly.

The definition of horizontal and vertical is not absolutely clear. Perhaps re-assess this bit of the text. Does it mean that you did not include cells that were neither vertical nor horizontal?

We categorized those PCs as ‘vertical’ that have a >30° angle relative to the PC layer, and those as ‘horizontal’ that have a <30° angle relative to the PC layer. All PCs are covered by these categories. This is now described on p. 5.